# WH2 and proline-rich domains of WASP-family proteins collaborate to accelerate actin filament elongation

Peter Bieling[1,2,3,*,†,‡] (iD), Scott D Hansen[1,†,§], Orkun Akin[1,¶], Tai-De Li[2,3,4,††], Carl C Hayden[5,‡‡], Daniel A Fletcher[2,3,4,**] (iD) & R Dyche Mullins[1,***] (iD)

## Abstract

WASP-family proteins are known to promote assembly of branched actin networks by stimulating the filament-nucleating activity of the Arp2/3 complex. Here, we show that WASP-family proteins also function as polymerases that accelerate elongation of uncapped actin filaments. When clustered on a surface, WASP-family proteins can drive branched actin networks to grow much faster than they could by direct incorporation of soluble monomers. This polymerase activity arises from the coordinated action of two regulatory sequences: (i) a WASP homology 2 (WH2) domain that binds actin, and (ii) a proline-rich sequence that binds profilin–actin complexes. In the absence of profilin, WH2 domains are sufficient to accelerate filament elongation, but in the presence of profilin, proline-rich sequences are required to support polymerase activity by (i) bringing polymerization-competent actin monomers in proximity to growing filament ends, and (ii) promoting shuttling of actin monomers from profilin–actin complexes onto nearby WH2 domains. Unoccupied WH2 domains transiently associate with free filament ends, preventing their growth and dynamically tethering the branched actin network to the WASP-family proteins that create it. Collaboration between WH2 and proline-rich sequences thus strikes a balance between filament growth and tethering. Our work expands the number of critical roles that WASP-family proteins play in the assembly of branched actin networks to at least three: (i) promoting dendritic nucleation; (ii) linking actin networks to membranes; and (iii) accelerating filament elongation.

**Keywords** actin; cytoskeleton; polymerase; profilin; WASP-family proteins
**Subject Categories** Cell Adhesion, Polarity & Cytoskeleton
**The EMBO Journal (2018) 37: 102–121**

## Introduction

Branched actin networks harness the free energy of actin filament assembly to generate forces required for many important cellular processes (Pollard & Cooper, 2009; Blanchoin *et al*, 2014). These self-assembling, cytoskeletal structures push against loads (generally cellular membranes) by promoting nucleation and elongation of actin filaments near the load surface (Pollard *et al*, 2000). Filament nucleation in branched networks is controlled by membrane-associated signaling molecules, which recruit nucleation-promoting factors (NPFs) that, in turn, localize the Arp2/3 complex and stimulate its actin nucleation activity (Pollard *et al*, 2000; Rotty *et al*, 2013). Filament elongation near the membrane surface is generally assumed to occur via diffusion-limited incorporation of actin monomers directly from solution (Pollard *et al*, 2000), with possible assistance from membrane-associated actin polymerases, such as formins and Ena/VASP proteins (Dominguez, 2009). The fact that neither formins nor Ena/VASP proteins are required for branched network formation (Bear *et al*, 2002; Block *et al*, 2012) has strengthened the idea that networks grow via interaction of soluble actin

1   Department of Cellular and Molecular Pharmacology and Howard Hughes Medical Institute, University of California, San Francisco, CA, USA
2   Department of Bioengineering & Biophysics Program, University of California, Berkeley, CA, USA
3   Chan Zuckerberg Biohub, San Francisco, CA, USA
4   Biological Systems & Engineering Division, Lawrence Berkeley National Laboratory, Berkeley, CA, USA
5   Sandia National Laboratories, Livermore, CA, USA
    *Corresponding author: Tel: +49 0231 133 2248; E-mail: peter.bieling@mpi-dortmund.mpg.de
    **Corresponding author: Tel: +1 510 664 4404; E-mail: fletch@berkeley.edu
    ***Corresponding author: Tel: +1 415 502 4838; E-mail: dyche.mullins@ucsf.edu
    † These authors contributed equally to this work
    ‡ Present address: Department of Systemic Cell Biology, Max Planck Institute of Molecular Physiology, Dortmund, Germany
    § Present address: Department of Chemistry and Biochemistry, University of Oregon, Eugene, OR, USA
    ¶ Present address: Department of Biological Chemistry, University of California, Los Angeles, CA, USA
    †† Present address: Advanced Science Research Center, City University of New York, New York, NY, USA
    ‡‡ Present address: Department of Biomedical Engineering, The University of Texas at Austin, Austin, TX, USA

monomers with free filament ends, and this has now become the textbook view (Phillips, 2013; Alberts, 2015; Pollard *et al*, 2017).

Mammalian cells express several distinct NPFs, including Scar/WAVE (Machesky *et al*, 1999), WASP (Derry *et al*, 1994), N-WASP (Rohatgi *et al*, 1999), WHAMM (Campellone *et al*, 2008), WASH (Liu *et al*, 2009), and JMY (Zuchero *et al*, 2009), each of which responds to unique cellular signals and contributes to distinct cellular processes (Campellone & Welch, 2010). These nucleation-promoting factors share a core set of conserved sequences, collectively known as a PWCA motif, that mediate interactions with profilin, actin, and the Arp2/3 complex. Their conservation across orthologous and paralogous proteins suggests that PWCA sequences function together as a unit, but important aspects of how the constituent sequence elements work together remain unclear.

The central/acidic (CA) sequence binds to the Arp2/3 complex (Marchand *et al*, 2001) and promotes a conformational change required for nucleation activity. The WASP homology 2 (WH2 or W) sequence (Husson *et al*, 2010; Renault *et al*, 2013; Dominguez, 2016) binds monomeric actin and promotes filament formation by delivering its actin payload to an Arp2/3 complex bound to the adjacent CA sequence (Marchand *et al*, 2001; Chereau *et al*, 2005). Efficient filament formation requires simultaneous association of an Arp2/3 complex with two actin monomer-loaded WCA sequences (Padrick *et al*, 2008, 2011; Ti *et al*, 2011; Boczkowska *et al*, 2014; Helgeson *et al*, 2014) on the side of a pre-existing (mother) filament (Mullins *et al*, 1998; Blanchoin *et al*, 2000). In addition to binding actin monomers and helping activate the Arp2/3 complex, WH2 domains also transiently interact with free barbed ends of actin filaments, effectively tethering filament networks to NPF-coated surfaces (Co *et al*, 2007). Other actin-binding proteins, such as Spire, Cobl, Ena/VASP, Thymosin-β4, Twinfilin, and ADF/cofilin also contain WH2-like sequences whose ability to bind actin contributes to a variety of activities such as nucleating, elongating, capping, and severing actin filaments as well as sequestering actin monomers (Dominguez, 2009; Husson *et al*, 2010; Renault *et al*, 2013).

In all known nucleation-promoting factors, a proline-rich domain (PRD) sits on the N-terminal side of the Arp2/3-activating WCA motif. These regions—whose function is not well understood—comprise multiple proline-rich sequences that bind either SH3-containing adaptor proteins or the actin monomer-binding protein profilin (Perelroizen *et al*, 1994; Petrella *et al*, 1996; Suetsugu *et al*, 1998). Profilin can bind both actin and poly-proline simultaneously, and, in the context of an NPF, this ternary complex promotes actin assembly in cell extracts (Yarar *et al*, 2002). The mechanism behind this effect has not been investigated in detail, but two models suggest themselves. Firstly, since profilin antagonizes binding of actin to WH2 domains, the proline-rich sequences might help shuttle actin from profilin–actin complexes onto the Arp2/3 complex. Secondly, the proline-rich domains might help promote filament elongation. This idea is supported by a few recent studies suggesting that WASP-family NPFs modulate actin filament elongation, but both stimulatory (Khanduja & Kuhn, 2014) and inhibitory (Sweeney *et al*, 2015) effects have been reported. Interestingly, filament elongation factors such as Ena/VASP proteins also contain proline-rich domains located on the N-terminal side of WH2-like sequences, and these proline-rich domains clearly contribute to actin polymerase activity (Hansen & Mullins, 2010).

To better understand how actin monomers incorporate into branched networks, we used *in vitro* reconstitution to investigate the effect of WASP-family proteins on growth of individual actin filaments and branched actin networks. We find that surfaces coated with WASP-family proteins potently enhance actin filament elongation in an Arp2/3-independent manner by recruiting a high density of polymerization-competent monomers to the surface, in proximity to growing filament ends. Interestingly, actin monomers bound to the WH2 domain and profilin–actin complexes bound to the PRD both contribute to enhanced filament elongation. Although profilin–actin complexes cannot interact directly with the WH2 domain, intramolecular (or *cis*) interactions promote the transfer of actin monomers from profilin–poly-proline complexes to the WH2 site. The rate enhancement caused by surface-associated NPFs implies that the majority of actin monomers enter a branched actin network via interaction with an NPF rather than directly from solution. On the other hand, when WH2 domains are not occupied by actin monomers, they can bind the barbed ends of nearby filaments, inhibiting elongation and tethering the network to the NPF-coated surface. Because of this bifunctional nature, the monomer occupancy of the WH2 domain helps to switch between the elongation-promoting and capping/tethering activities of NPF proteins. Our work reveals that, in addition to promoting Arp2/3-dependent filament nucleation, high local densities of WASP-family proteins also collectively function as distributive *network polymerases* and transient tethering factors, and might be more accurately called "network promoting factors".

## Results

### Filament elongation from soluble monomers is not fast enough to account for the rate of branched actin network growth

To better understand the process of filament elongation within branched actin networks, we compared the growth of these networks to the elongation of single actin filaments in the same concentration of profilin–actin complexes. We used time-lapse total internal reflection fluorescence (TIRF) microscopy to measure elongation of single actin filaments growing from surface-tethered [via biotin-PEG/streptavidin/biotin-heavy-meromyosin linkage (Hansen *et al*, 2013) and see Materials and Methods] and phalloidin-stabilized actin filament seeds (Fig 1A and B). In the presence of 5 μM profilin–actin complexes, filaments elongated exclusively from their fast-growing, barbed end at a rate of 3.6 ± 0.4 μm/min. This elongation rate agrees with our previous observation (Hansen & Mullins, 2010) that profilin slows barbed end growth of non-muscle actin filaments by 55% (Fig EV1A and B), a much larger effect than previously observed in experiments using rabbit skeletal muscle actin (Kang *et al*, 1999, also see Materials and Methods). We next measured the velocity of branched actin networks, composed of many short, Arp2/3-nucleated filaments, also growing in 5 μM profilin–actin. For this assay, we first coupled an Arp2/3 activating PWCA fragment of WAVE1 (274–559aa) at high densities (about 15,000 molecules/μm², see Materials and Methods) to lipid-coated beads, via interaction between an N-terminal His$_{10}$ sequence in the polypeptide and Ni-conjugated lipids doped into the bilayer. We then initiated actin network assembly by mixing these lipid- and

protein-coated particles with the Arp2/3 complex, heterodimeric capping protein, and profilin–actin (Fig 1C and D). When we combined 5 μM profilin–actin with concentrations of capping protein (200 nM) and Arp2/3 complex (50 nM) previously shown to promote rapid network growth (Loisel *et al*, 1999; Akin & Mullins, 2008), we created branched actin networks that elongated at a rate of 7.2 ± 1.48 μm/min, more than twice the speed of individual filaments grown in the same concentration of profilin–actin. From previous work (Schafer *et al*, 1996; Blanchoin *et al*, 2000), we know that neither capping protein nor the Arp2/3 complex accelerates elongation of individual filaments. The balance of capping and nucleation activity, however, does influence the growth velocity of dendritic networks by regulating the interaction of barbed ends with WASP-family proteins (Bernheim-Groswasser *et al*, 2002; Akin & Mullins, 2008; Bieling *et al*, 2016).

The convex surface of NPF-coated microspheres creates internal stresses in the dendritic actin network (Bernheim-Groswasser *et al*, 2002; Dayel *et al*, 2009) and directs assembly of many filaments that do not point in the direction of motion. Both of these effects reduce the growth velocity of networks assembled from spherical particles. To overcome these confounding effects of convex surfaces, we assembled branched networks from flat glass coverslips micropatterned with the Arp2/3-activating domain of WAVE1 (Fig 1E and F; Fourniol *et al*, 2014; Bieling *et al*, 2016). The density of NPFs on the micropatterned surfaces (~2,000/μm$^2$, see below) is lower than that on the lipid-coated beads, but their geometry more closely approaches the concave inner surface of a lamellipodial protrusion. In addition, this geometry decreases internal stress in the network and produces more filaments aligned with the direction of net growth. Branched actin networks assembled from these planar, micropatterned surfaces in 5 μM profilin–actin grew at 12.6 ± 1.2 μm/min, significantly faster than networks assembled on beads and approximately 3.5-fold faster than individual filaments (Fig 1G and H).

### WASP-family proteins function as distributive actin polymerases

Formally, the accelerated growth of branched networks could be either (i) a meso-scale property that emerges from the collective behavior of a large number of growing filaments (E. Hohlfeld & P. Geissler, personal communication) or (ii) a micro-scale, biochemical effect of WASP-family proteins on individual filament assembly. To test the micro-scale effects, we imaged individual actin filaments polymerizing in the presence and absence of WASP-family proteins attached to a planar surface (Fig 2A). We attached actin filaments via a biotin-PEG/streptavidin/biotin-heavy-meromyosin linkage to

coverslips micropatterned with squares (14 × 14 μm) of WAVE1, and we drove elongation of the surface-associated filaments with 1 μM profilin–actin. We observed growing filaments by TIRF microscopy but, because the immobilized WAVE1 molecules recruit actin monomers and profilin–actin complexes to the coverslip surface, we could not use fluorescently labeled actin to monitor filament growth (Fig 2B). Instead, we added low concentrations of an actin filament-binding domain from utrophin, conjugated to a fluorescent dye (Alexa488-UTRN$_N$; Burkel *et al*, 2007). When used at low concentrations (5–10 nM), the utrophin-based probe accurately reports the length of actin filaments without affecting their rate of polymerization (Fig EV1C and D), enabling us to visualize filaments growing against a high-density background of WAVE1-bound actin monomers (Fig 2C).

We observed a dramatic difference between the rate at which filaments grow across the passivated glass surface and the rate at which they grow across squares of immobilized WAVE1. Specifically, barbed ends located within a WAVE1 square elongate almost threefold faster than barbed ends outside a WAVE1 square, from 1.51 ± 0.20 μm/min to 4.15 ± 0.65 μm/min (Fig 2C and E, Movie EV1). In addition, whenever a growing barbed end crossed the boundary of a WAVE1 square, its growth velocity instantaneously increased or decreased, depending on whether it crossed into or out of the high-density WAVE1 region (Fig 2D, Movie EV1). To account for the potential influence of residual WAVE1 molecules located outside of the high-density WAVE1 pattern (due to imperfect photolithographic patterning, see Materials and Methods), we also performed independent control experiments on passivated but non-patterned surfaces (Fig 2E, "negative control"), revealing a barbed end growth rate of 0.71 ± 0.06 μm/min, consistent with a solution concentration of 1 μM profilin–actin (Fig EV1B). These experiments revealed that high local densities of WAVE1 accelerate polymerization by about sixfold (Fig 2E) and that even low NPF densities outside the passivated regions (~10% of maximum mCherry-WAVE1 density) could markedly accelerate actin filament elongation (Fig EV2A).

To characterize the relationship between NPF density and filaments elongation in greater detail, we first determined the absolute surface density of mCherry-WAVE1 molecules on PEG-functionalized surfaces under our experimental conditions by spatial fluctuation analysis (see Materials and Methods). This revealed that our immobilization method resulted in densities of about 1,850 ± 500 molecules/μm$^2$, which approximately matches physiological NPF densities at sites of actin assembly (Footer *et al*, 2008; Arasada & Pollard, 2011; Ditlev *et al*, 2012). We then systematically reduced the density of immobilized mCherry-WAVE1 by spiking in a non-fluorescent mutant mCherry as a competitor for

---

**Figure 1. Dendritic actin networks can grow faster than individual filaments under the same biochemical conditions.**

A Scheme of individual actin filaments polymerizing on functionalized glass surfaces from profilin–actin in solution.

B Time-lapse TIRF microscopy of single-filament growth from Alexa488 phalloidin-stabilized seeds (green) in the presence of 5 μM profilin–actin.

C Scheme of branched actin filaments polymerizing within dendritic actin networks assembled from NPF- and lipid-coated microspheres with profilin–actin, CP, and Arp2/3 in solution.

D Wide-field epifluorescence images of indicated network components of bead-attached dendritic networks after kinetic arrest at indicated times. Biochemical conditions are matching (B) with the exception of the additional presence of 200 nM CP and 50 nM Arp2/3.

E Scheme of dendritic network growth from square WAVE1-micropatterns on a two-dimensional coverslip with profilin–actin, CP, and Arp2/3 in solution.

F Reconstructed axial view for indicated network components from confocal imaging of dendritic network growth. Conditions are as in (D).

G Average length of individual filaments (red, *N* = 15), bead- (green, *N* = 28–48 per time point) or NPF micropattern-assembled (blue, *N* = 20) dendritic networks as a function of time. Error bars represent SD.

H Distributions of growth velocities for individual filaments (red, *N* = 125), bead- (green, *N* = 191) or NPF micropattern- (blue, *N* = 65) assembled dendritic networks.

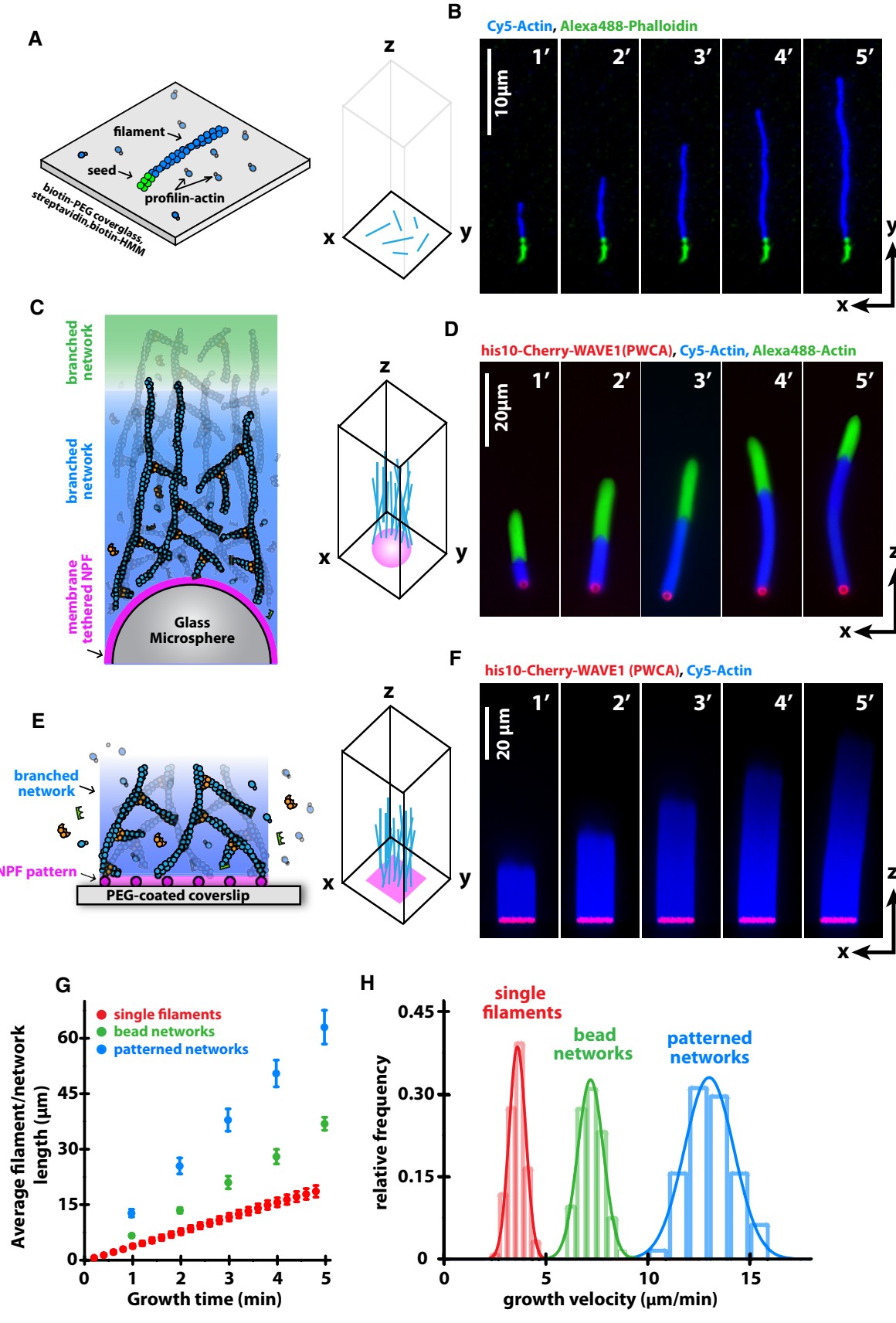

**Figure 1.**

immobilization and performed single-filament TIRF assays on these surfaces. We observed that the barbed end growth velocity decreased approximately linearly with the drop in measured mCherry-WAVE1 surface density over the whole experimentally accessible range (Fig 2F).

In contrast to a recent report (Khanduja & Kuhn, 2014), we did not detect persistent attachment of growing barbed ends and/or buckling of filaments within the NPF patches, characteristic for processive polymerases such as formin- or Ena/VASP-family proteins (Kovar & Pollard, 2004). Instead, filament barbed ends elongate at constant velocity across the WAVE1-coated surface even at the highest NPF surface densities (Fig 2C and D). In separate, single-molecule experiments we did not detect binding of soluble, fluorescently labeled constructs of WAVE1 to growing barbed ends, further supporting the idea that the interaction between growing filament ends and WAVE1 is extremely short-lived (Fig EV2B and C).

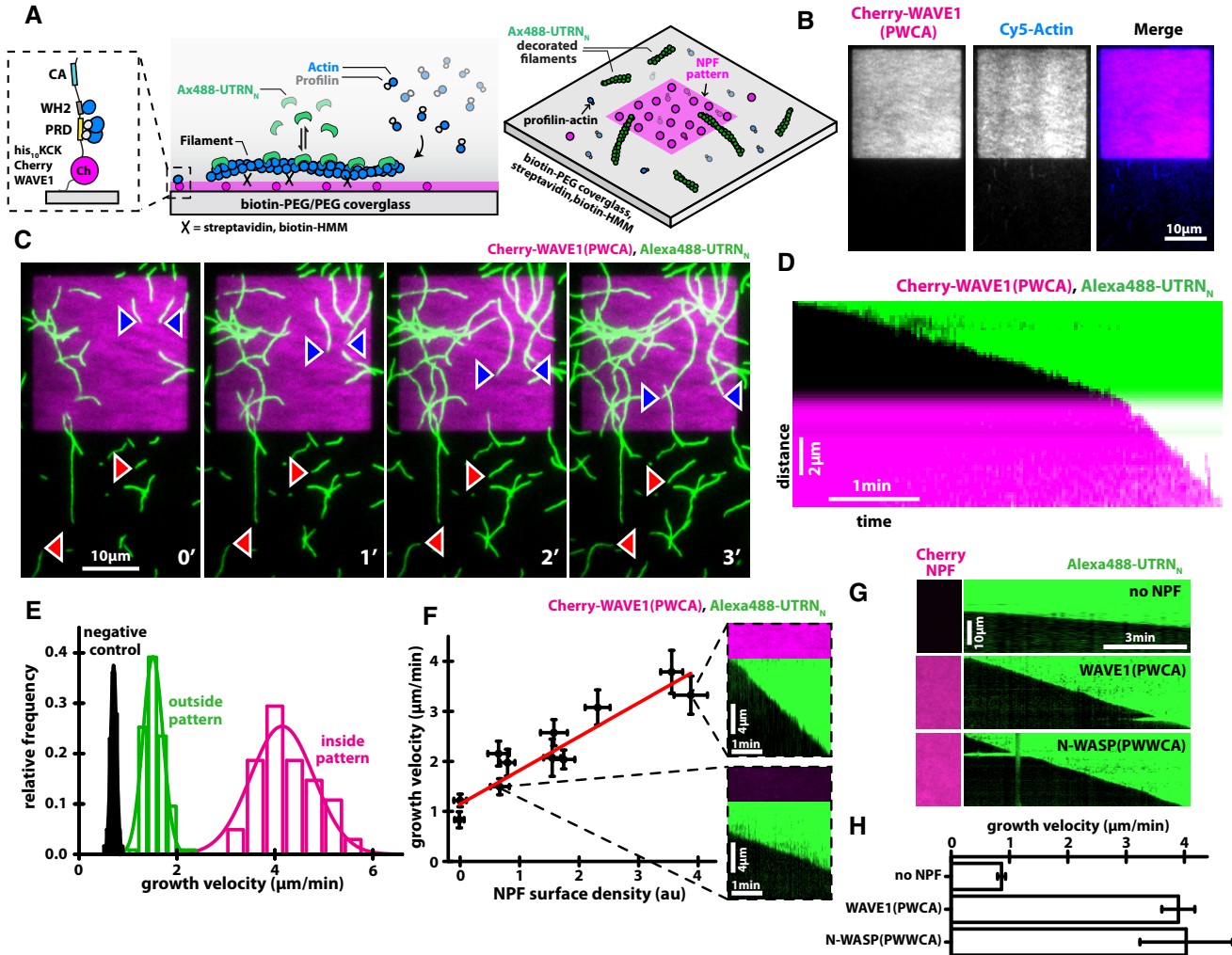

**Figure 2. NPFs promote filament elongation locally.**

A   Scheme of actin filaments growing on functionalized glass surfaces containing WAVE1 micropatterns in the presence of profilin–actin in solution.

B   TIRF microscopy of actin monomers (blue, 1 μM total, 30% Cy5 labeled) in the presence of equimolar profilin recruited to a WAVE1 pattern (magenta).

C   TIRF microscopy of actin polymerization in- (blue arrows) and outside (red arrows) of WAVE1-micropatterns (magenta) in the presence of 1 μM profilin–actin visualized by a filament-binding probe (5 nM Alexa488-UTRN$_N$).

D   Kymograph of a growing filament barbed end entering a WAVE1 micropattern. Conditions as in (C).

E   Distributions of barbed end growth velocities for filaments growing outside (green, $N = 115$) or inside (magenta, $N = 102$) of WAVE1-micropatterns compared to a negative control (non-NPF-treated PEG surface, black, $N = 61$).

F   Left: Barbed end growth velocities (dots) in 1 μM profilin–actin as a function of measured mCherry-WAVE1 surface density. NPF surface density was varied using varying amounts of dark mCherry competitor (see Materials and Methods). The red line is a linear fit to the data. Error bars are SD. Right: Surface mCherry-WAVE1 fluorescence and corresponding kymographs of barbed end growth for indicated examples. $N = 12$ experiments, > filaments per experiments.

G   Surface mCherry-NPF fluorescence and corresponding kymographs of barbed end growth for indicated WAVE1, N-WASP, or dark mCherry control constructs in 1 μM profilin–actin.

H   Barbed end growth velocities for indicated WAVE1, N-WASP, or dark mCherry control constructs in 1 μM profilin–actin. Error bars represent SD. At least 50 individual filaments from two independent experiments were analyzed per condition.

In contrast to the behavior of weakly processive Ena/VASP polymerases attached to beads (Breitsprecher *et al*, 2008), biotinylated WAVE1 conjugated to streptavidin-functionalized polystyrene beads did not support processive attachment and elongation of individual actin filaments (Fig EV2D). Taken together, these experiments show that surface-associated NPF molecules can potently accelerate elongation of nearby barbed ends, an activity that does not involve or require processive attachment of individual actin filaments.

Finally, we investigated whether polymerase activity is a conserved property shared by other WASP-family proteins. We observed that surface-immobilized N-WASP fragments containing the core PWWCA motif were similarly potent as WAVE1 in enhancing the elongation of nearby barbed ends growing in 1 μM profilin–actin (Fig 2G and H). In summary, these experiments demonstrate that WASP-family proteins not only promote nucleation activity of the Arp2/3 complex, but, when concentrated on a surface, can function as distributive polymerases that accelerate actin filament elongation.

### The WH2 domain of WAVE1 accelerates actin filament elongation in the absence of profilin

Every WASP-family protein contains one or more actin monomer-binding WH2 domains, required to promote actin filament nucleation by the Arp2/3 complex (Dominguez, 2016). Similar to profilin–actin complexes, WH2–actin complexes readily interact with the barbed end of existing filaments (Higgs *et al*, 1999), suggesting that they might confer polymerase activity to WASP-family proteins (Fig 3A). To test this idea, we monitored the growth of actin filaments from seeds immobilized on surfaces coated with various WAVE1 protein constructs (Fig 3B). In 1 μM monomeric actin, a construct composed of mCherry fused to the C-terminal region of WAVE1—comprising the proline-rich, WH2, and CA sequences (PWCA construct, Fig 3A–C)—recruited actin monomers to the patterned surface and accelerated elongation of surface-proximal filaments by 3.5-fold (compared to immobilized mCherry; Fig 3B and C). Molecular dissection of this WAVE1 construct revealed that the WH2 domain was both necessary and sufficient for this acceleration effect. WAVE variants lacking the WH2 sequence (P_CA constructs) failed to recruit actin monomers and accelerate filament growth (Fig 3B and C), while a construct containing *only* the WH2 sequence (W construct) both recruited monomeric actin and promoted filament growth (Fig 3B and C). To determine the contribution of local WAVE1 density to the polymerase activity of WH2 domains, we compared barbed end elongation rates from surface-immobilized seeds in the presence or absence of WAVE1 WCA in solution

(Fig 3D and E). Unlike surface-immobilized WAVE1 constructs, soluble WAVE1 added at a 1:1 stoichiometry to actin actually inhibited filament growth (Fig 3E). A simple quantitative model, based on the assumption that actin and actin-WH2 complexes bind barbed ends with different rate constants, entirely explains this effect (Fig 3F, see Materials and Methods). Fitting this model to our experimental data suggests that WH2–actin complexes associate with free barbed ends about fivefold more slowly than free actin monomers (Fig 3F). We also confirmed that WH2 binding does not significantly alter the dissociation rate of ATP-bound monomers from the barbed ends (Fig EV2), further supporting that the inhibitory effect originates from reduced association. These data reveal that, although soluble WAVE1 WH2 domains slow actin filament elongation [consistent with previous observations by Sweeney *et al* (2015)], the same WH2 domains immobilized at high density on a surface actually accelerate filament growth by providing a high local density of polymerization-competent WH2-bound actin monomers.

### The WAVE1 WH2 domain caps and tethers filament barbed ends in the absence of monomeric actin

The surface-immobilized WH2 domain from WAVE1 accelerates incorporation of actin monomers into growing filaments, but how does this activity relate to the more physiologically relevant process of assembling a branched network from profilin–actin complexes? To tackle this question, we reconstituted actin network assembly from lipid-coated beads containing N-terminal His$_{10}$-tagged proteins tethered to Ni-NTA lipids. We attached a constant 3:2 ratio of intact WAVE1 PWCA and either an inactive protein (i.e. mCherry) or a different fragment of WAVE1 to the bilayer (Fig 4A). We used the WAVE1 PWCA construct in these experiments because an intact proline-rich domain is required for efficient actin network formation in the presence of profilin (Fig EV3) and we chose a PWCA density sufficient to drive Arp2/3-dependent network assembly. We also verified that none of the WAVE1 fragments other than PWCA could autonomously activate the Arp2/3 complex and construct a dendritic actin network (Fig EV3). Dendritic actin network assembly was initiated by adding Arp2/3 (50 nM), capping protein (200 nM), and profilin–actin complexes (5 μM). In contrast to the previous single-filament assays in actin alone (Fig 3B and C), WH2 domains distributed *in trans* among the WAVE1 PWCA constructs on the microsphere surface significantly slowed the growth of branched actin networks (Fig 4B and C).

WH2 domains in every known WASP-family protein lie proximal to proline-rich domains (PRDs) capable of binding profilin–actin complexes (Perelroizen *et al*, 1994; Petrella *et al*, 1996; Suetsugu

---

**Figure 3. NPFs can stimulate filament growth by recruiting monomers to their WH2 domain.**

A    Domain architecture and boundaries for WAVE1 fragments used in this study.

B    Scheme of domain architecture of surface-immobilized WAVE1 variants (top panel), time averages (middle panels, 7 min total), and representative kymographs (lower panel) from multicolor TIRF imaging of actin polymerization on WAVE1-coated coverslips. Conditions are 1 μM actin, 5 nM Alexa488-UTRN$_N$.

C    Average barbed end growth velocities of actin polymerization on WAVE1-immobilized surfaces. At least 40 filaments were analyzed per condition. Error bars represent SD.

D    Scheme of actin polymerization from NPF–actin complexes in solution.

E    TIRF microscopy of actin polymerization from 2 μM actin in the absence (upper panel) or presence of near-saturating amounts of WAVE1 WCA.

F    Barbed end polymerization rates as a function of actin and WAVE1-actin ratios. The red line is a fit to a minimal kinetic model assuming two polymerization-competent actin species that associate to filament ends with different on-rates as indicated (see Materials and Methods). Error bars represent SD. At least 40 filaments were analyzed per condition.

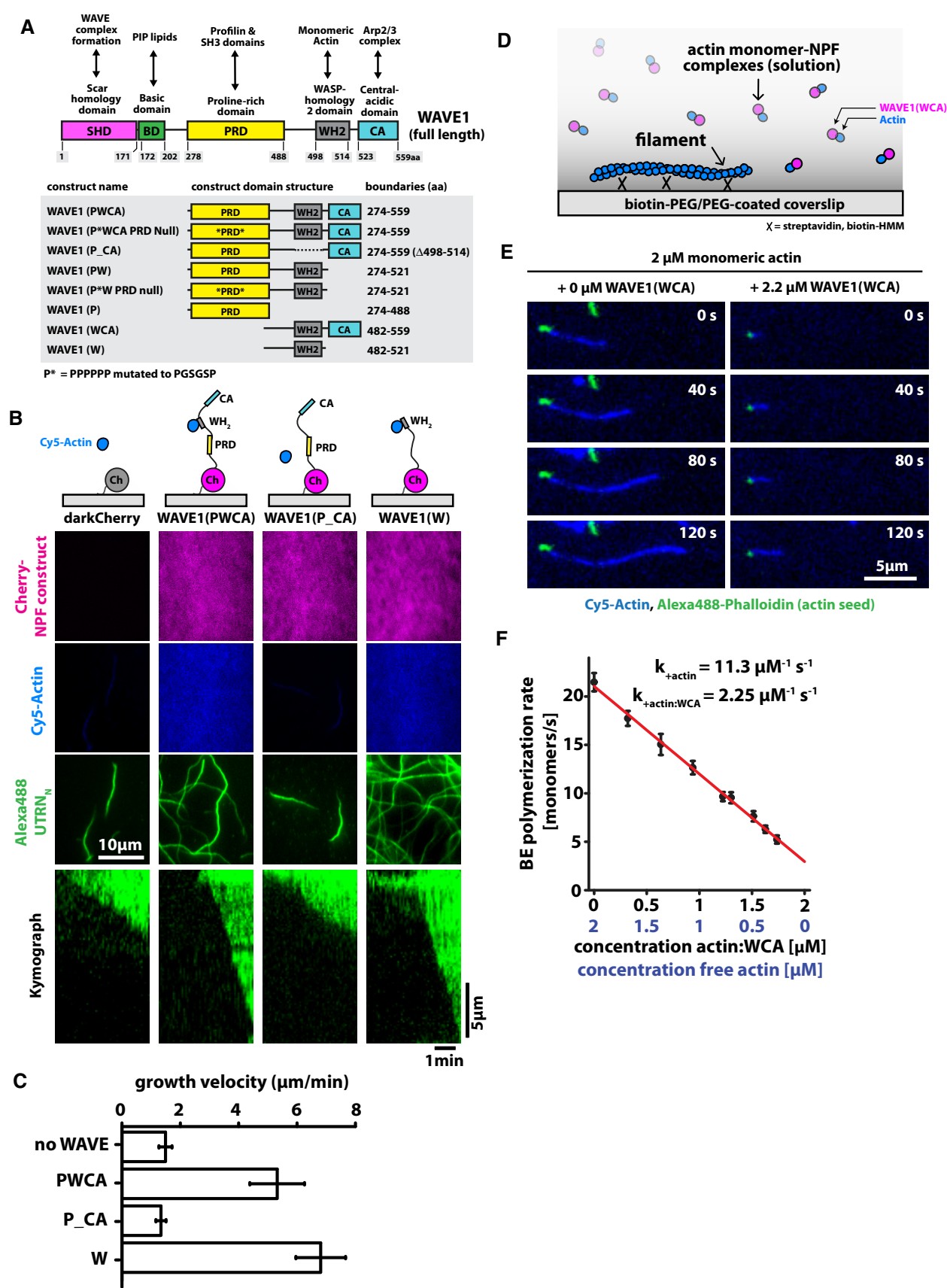

Figure 3.

*et al*, 1998). Formally, these proline-rich regions could have several different functions, including (i) enhancing filament nucleation by supplying actin to the Arp2/3 complex; (ii) accelerating filament elongation by providing actin to nearby filament ends; and (iii) shuttling actin from profilin–actin complexes onto adjacent WH2 domains. We ruled out the first hypothesis by demonstrating that the proline-rich regions cannot rescue Arp2/3 activation by constructs lacking a functional WH2 domain, even in the presence of profilin (Fig EV3). To test the alternative hypotheses, we co-mingled WAVE1 PWCA domains with constructs containing either the proline-rich and WH2 domains (PW construct) or the proline-rich region alone (P construct). We tested the ability of these mixtures to generate branched actin networks. Decisively, addition of any supplemental construct containing a wild-type proline-rich domain enhanced actin network growth (Fig 4B and C). This enhancement depended on the ability of the proline-rich domain to bind profilin, as demonstrated by the effect of a mutation that abolishes profilin binding (PW PRD null, Fig EV4A). Notably, even the P construct (containing only the proline-rich domain) enhances network growth, supporting the notion that proline-rich sequences can autonomously deliver profilin–actin complexes to nearby growing filament ends.

The density and velocity of branched actin networks react to changes in a variety of physical and biochemical parameters (Loisel *et al*, 1999; Bernheim-Groswasser *et al*, 2002; Akin & Mullins, 2008; Bieling *et al*, 2016). Therefore, to more directly probe the polymerase activity of WASP-family proteins in the presence of profilin, we returned to TIRF microscopy studies of individual actin filaments growing across WAVE1-coated glass surfaces (Fig 4D and E). Similar to its effect on branched network growth, the isolated WAVE1 WH2 domain potently inhibited filament elongation by profilin–actin, while PW constructs that contain wild-type PRD and WH2 domains dramatically accelerated elongation. Mutant proline-rich domains defective in profilin binding, in contrast, completely failed to confer polymerase activity (Fig 4D and E). These results indicate the PRD alone can confer polymerase activity in the presence of profilin–actin and that tandem PRD and WH2 domains function synergistically to enhance polymerase activity.

Our branched network and single-filament experiments with WAVE1 demonstrate that the WH2 and proline-rich domains are both potent actin polymerases, with the first requiring free actin as a substrate and the second requiring profilin–actin. The intermediate polymerization rates observed when both sequences are present in the same polypeptide could reflect capping of barbed ends by unoccupied WH2 domains and/or competition between proximal

WH2 domains and nearby barbed ends for the actin bound to profilin:poly-proline complexes. Operation of the second mechanism would imply that the proline-rich domain can transfer actin from profilin–actin complexes onto the WH2 domain.

### Transfer of actin from profilin:poly-proline complexes to adjacent WH2 domains

The amino acid sequence of the proline-rich region of WAVE1 contains six putative profilin-binding sites (Fig EV4A). To characterize the interaction between profilin and the PRD, we constructed a series of mutants in which we ablated all (PRD null), or all but one, of the potential profilin-binding sites. By mixing each PRD mutant construct with an excess of profilin and performing sedimentation equilibrium ultracentrifugation (Fig EV4B and C), we identified two primary, high-affinity binding sites [Site B (322–332aa) and Site F (424–435aa), $K_d$~2 μM and 4 μM] along with four lower affinity sites ($K_d$ > 10 μM, Fig EV4D). To verify the ability of profilin:actin complexes to bind these sites, we performed sedimentation equilibrium on two WAVE1 PRD variants (i.e., PRD [B] and [F]), each with a single high-affinity profilin-binding site intact. We mixed each PRD variant with Latrunculin B-stabilized actin monomers and various concentrations of profilin (Fig 5A). Increasing the concentration of profilin in these experiments significantly increased the apparent mass of both PRD variants (Fig 5A). Consistent with formation of a ternary complex between WAVE1, profilin, and actin, the observed change in mass of each PRD variant was more than twice the mass of profilin alone.

Finally, we investigated the biochemical crosstalk between adjacent PRD and WH2 domains. Profilin and the WH2 domain bind to partially overlapping sites on an actin monomer (Chereau *et al*, 2005), and so any potential ternary complex formed between these three molecules must be short-lived. We hypothesized that a proline-rich domain in close proximity to a WH2 sequence might significantly increase the local concentration of profilin:actin, thereby facilitating transfer of actin to the WH2 domain. To test this hypothesis, we developed a Förster resonance energy transfer (FRET) method to directly measure actin monomer occupancy of the WAVE1 WH2 domain (Fig 5B). Briefly, we conjugated a fluorescence donor (Alexa488) to a single cysteine residue on the N-terminal side of WH2 domains in both WAVE1 PWCA and WAVE1 WCA constructs. We then chose a non-fluorescent acceptor (Atto540Q) with an absorption spectrum matched to the emission of the donor and conjugated it to monomeric actin. The two labels had little impact (≤ twofold) on the affinity of actin-WH2 and actin–profilin

---

**Figure 4. Profilin shunts NPF WH2 domains to barbed ends where they inhibit filament growth.**

A   Scheme of the domain architecture of WAVE1 variants immobilized to lipid-coated beads. Nucleation is always driven from wild-type WAVE PWCA (60% total), while other variants (40% total) do not activate the Arp2/3 complex (Fig EV4A and B) but can affect filament elongation.

B   Wide-field epifluorescence images (Alexa488-actin) of bead-attached dendritic networks following kinetic arrest after 10 min of network growth for spike-in experiments (i.e., 60% his10-darkCherry-PWCA, 40% his10-Cherry-W; molar percentage) using indicated WAVE1 variants. Conditions are 5 μM profilin–actin, 200 nM CP, and 50 nM Arp2/3. For all conditions, actin network assembly was quenched by adding 5-fold molar excess Latrunculin B and phalloidin 10 min after initiating the reaction.

C   Average network growth rates for indicated WAVE1 variants or mCherry as a negative control (*N* = 57–73 beads per condition). Error bars represent SD.

D   Scheme of domain architecture of surface-immobilized WAVE1 variants (top panel), time averages (middle panels, 7 min total), and representative kymographs (lower panel) from multicolor TIRF imaging of polymerization from profilin–actin on WAVE1-coated coverslips. Conditions are 2 μM actin, 3 μM profilin, 5 nM Alexa488-UTRN$_N$.

E   Average barbed end polymerization rates for indicated WAVE1 variants. Error bars represent SD. At least 40 filaments were analyzed per condition.

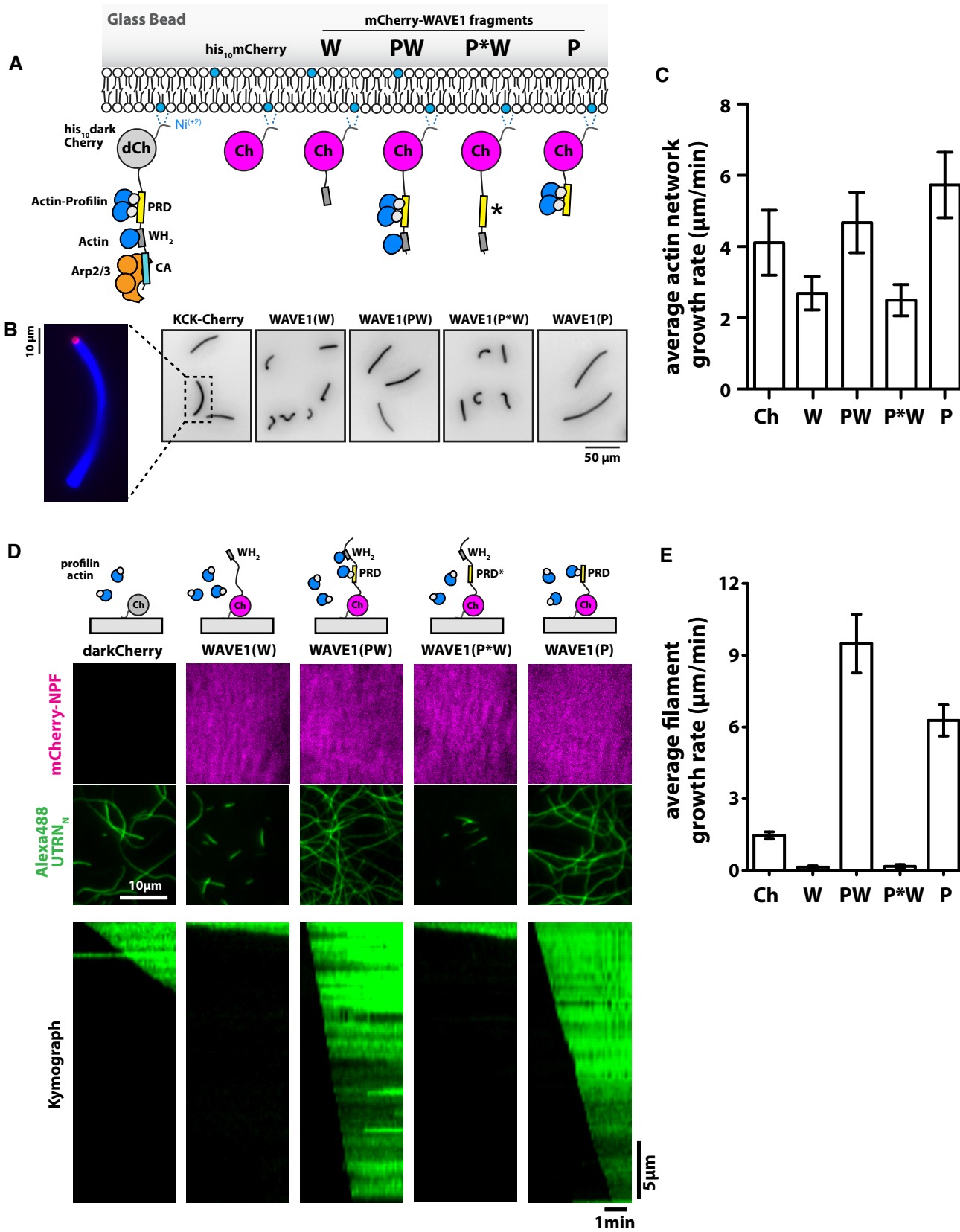

**Figure 4.**

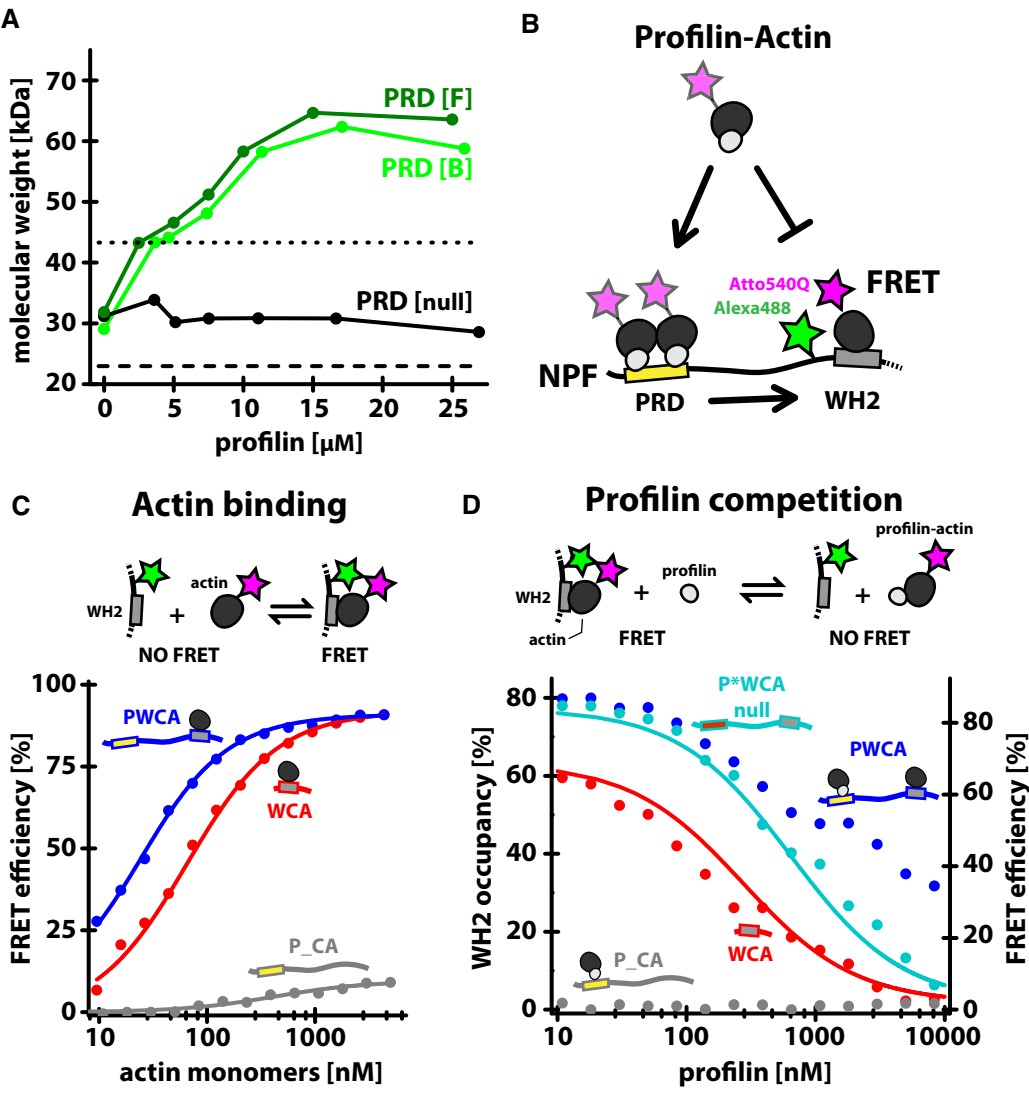

**Figure 5.  Monomer shuttling from the PRD regulates the monomer occupancy of the WH2 domain.**

A   The weighted mass average of WAVE1 PRD (15 μM) determined by sedimentation equilibrium ultracentrifugation with LatB-stabilized monomers (15 μM) and various profilin concentrations as indicated. The lower dashed line indicates the expected mass of WAVE1 PRD; the 5–7 kDa upshift is due to non-specific actin binding (WAVE1 PRD[null], black symbols). The upper dashed line marks the equilibrium mass observed for 15 μM WAVE1 PRD[B] in the presence of 90 μM profilin alone.

B   Experimental scheme of detecting WH2 occupancy by FRET.

C   FRET efficiencies for donor (Alexa488)-labeled WAVE1 variants (c = 7.5 nM, PWCA = blue, WCA = red, P_CA = gray) as a function of LatB-stabilized, quencher (Atto540Q)-labeled actin monomers concentration. Lines are fits to a single site binding curve.

D   WH2 occupancies (left axis, Materials and Methods) and FRET efficiencies (right axis) for donor (Alexa488)-labeled WAVE1 variants (c = 7.5 nM, PWCA = blue, WCA = red, P*WCA(null) = cyan, P_CA = gray) as a function of profilin concentration. Concentration of LatB-stabilized, Atto540Q-actin monomers was constant (175 nM) at all conditions. Lines are fits to a competitive inhibition model (see Materials and Methods).

interactions as determined by fluorescence anisotropy- or FRET-based binding assays (Fig EV5). Acceptor-labeled actin strongly reduced (92% efficiency) the fluorescence of donor-labeled WAVE1 constructs in a dose-dependent manner (Fig 5C). Quenching was specific to WH2 binding, since a WH2-delete construct (P_CA) showed negligible quenching at saturating actin concentrations (9% efficiency), most likely caused by weak binding ($K_d = 425$ nM) to the central (C) domain distal to the donor (Kelly *et al*, 2006). FRET-based binding assays also revealed that actin bound with high affinity to the WH2 domain of both WCA and PWCA ($K_d = 61$ and

21 nM, respectively, Fig EV5D) with the larger construct binding somewhat more tightly (Fig 5C). We then formed the binary actin:WH2 complex, using either the WCA or PWCA construct, and titrated increasing concentrations of profilin (Fig 5D) in FRET-based competition experiments. Consistent with previous studies (Heimsath & Higgs, 2012; Suarez *et al*, 2015), we found that addition of profilin to the shorter WCA construct, lacking the PRD, produced a dose-dependent release of quencher-labeled actin from the WH2 domain and, at saturating profilin levels, resulted in full recovery of donor fluorescence, suggesting complete dissociation of

acceptor-labeled actin. These data reveal little evidence for a ternary complex and fit a simple binding model in which profilin and the WCA construct directly compete for actin monomers. We obtained the same result (Fig 5D) using a PWCA construct deficient in profilin binding (PRD null). In contrast, a PWCA construct competent to bind profilin exhibited a significant degree of residual fluorescence quenching (> 30%), even at highest profilin concentrations used (Fig 5D). This residual FRET signal depended on the presence of a functional WH2 domain, because we observed no fluorescence quenching—even at high profilin concentrations—in a WH2-delete construct (P_CA). These results are incompatible with a simple competition model and reveal that a significant fraction (> 35%) of WH2 domains contact the actin recruited to a PWCA construct via the interaction of profilin:actin with a nearby proline-rich region. In conclusion, our data reveal that profilin can antagonize actin binding to the WAVE1 WH2 domain but that nearby proline-rich regions partially alleviate this inhibition by providing a high effective concentration of profilin–actin complexes.

## Discussion

### WASP-family proteins as distributive network polymerases

The growth of branched actin networks is a complex, multistep process that requires continual nucleation, elongation, and capping of actin filaments, most of which do not lie exactly parallel to the direction of network growth. Despite this biochemical and architectural complexity, we find that the growth velocity of branched actin networks can exceed the diffusion-limited elongation of isolated actin filaments in the same concentration of polymerizable actin. From the surprisingly fast growth of branched actin networks, we discovered that surface-associated WASP-family proteins promote rapid elongation of nearby actin filaments by providing a high-density source of polymerization-competent monomers (Fig 6). Importantly, the acceleration of filament elongation must originate from the transfer of monomers directly from NPF-bound pool (Fig 6C, Model 1) and cannot be the result of WASP-family proteins increasing the proximal concentration of free, monomeric actin in solution (Fig 6C, Model 2). The latter mechanism would be incompatible with thermodynamic laws because, at steady state, the rates of binding and unbinding of monomers to the NPF surface must balance.

Polymerase activity of WASP family was not previously detected in bulk assays performed using soluble NPFs (Higgs *et al*, 1999; Marchand *et al*, 2001; Sweeney *et al*, 2015), which actually slow polymer formation by (i) preventing spontaneous filament nucleation and (ii) reducing the association kinetics of bound monomers to barbed ends (Fig 3F). To generate functional branched actin networks, however, these proteins must be concentrated on two-dimensional surfaces. Only by reconstituting these physiologically relevant boundary conditions did we observe the strong polymerase activity intrinsic to WASP-family proteins. This effect on filament

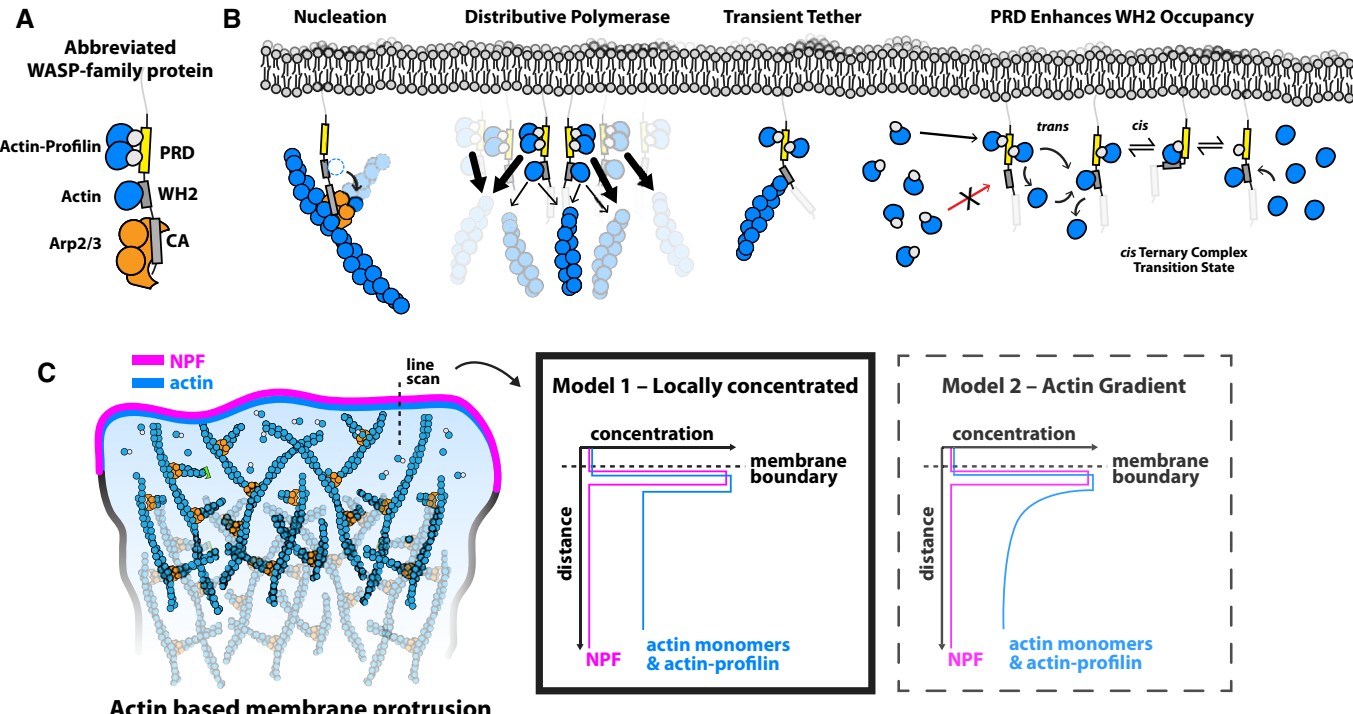

**Figure 6. WASP-family proteins are network promoting factors.**

A Scheme of architecture and binding partners of the WAVE1 PWCA core domains.
B Overview of biochemical functions of WASP-family proteins in dendritic network assembly.
C Two alternative models for the polymerase activity of WASP-family proteins. Note that the second model is not in agreement with thermodynamic laws (see Discussion).

elongation rivals that of dedicated actin polymerases such as formins and Ena/VASP proteins but, since WASP-family proteins also promote nucleation activity of the Arp2/3 complex, the key substrates of their polymerase activity are the branched actin networks that they help create. We therefore refer to the filament growth promoting activity of WASP-family proteins as a *network polymerase* activity.

Although concentrating WASP-family proteins on a surface can create an effective network polymerase, individual WH2-associated actin monomers are relatively poor polymerization substrates. We do not know whether this effect of the WH2 domain is steric or allosteric (Chereau *et al*, 2005), but we speculate that it might be important for tuning the bias between filament elongation and Arp2/3-dependent nucleation. More generally, actin-binding WH2 domains are found in many different proteins and each might be tuned to funnel actin monomers to a preferred downstream partner.

Formin-family actin polymerases maintain long-lived contact with the terminal monomers of a growing filament (Kovar *et al*, 2006). Their ability to accelerate filament assembly relies on a proline-rich sequence that recruits multiple profilin–actin monomers to the vicinity of the growing filament end (Paul & Pollard, 2009). Ena/VASP polymerases are tetrameric proteins with distinct sequences that bind actin filaments, monomers, and profilin–actin complexes. The interaction between an individual Ena/VASP polypeptide and a growing actin filament is short-lived, but the processivity of these transient polymerases increases dramatically with oligomerization and higher-order clustering (Breitsprecher *et al*, 2008, 2011; Hansen & Mullins, 2015). WASP-family proteins, on the other hand, cannot support processive polymerase activity at the level of individual filaments, even when clustered on surfaces (Fig EV2D), because they are monomeric and lack separate actin monomer and filament-binding domains. In contrast to a previous study that employed non-specific adsorption of artificially dimerized N-WASP fragments to microneedles (Khanduja & Kuhn, 2014) and earlier theoretical "actoclampin" models of filament elongation (Dickinson & Purich, 2002), we show that processive attachment is in fact not *required* for the enhancement of filament elongation by NPFs. This type of distributive polymerase activity, which arises from the collective action of many surface-immobilized WASP molecules, is likely to be a feature of other actin monomer-binding proteins concentrated at sites of actin assembly. For example, monomeric, non-processive Ena/VASP mutant proteins can also accelerate local actin filament assembly when concentrated on surfaces (Breitsprecher *et al*, 2008; Brühmann *et al*, 2017).

This polymerase activity of WASP-family proteins alters our "textbook" picture of branched actin network assembly, which usually portrays actin monomers adding to growing filament ends directly from solution. A simple calculation, however, reveals that more than 80% of the actin in our reconstituted networks is delivered to the filaments directly from the WASP-family proteins on the surface of the load. Briefly, if we assume that soluble and NPF-associated actin monomers have similar access to free barbed ends, the fraction of actin subunits that enter the network from solution depends on the relative rates of incorporation from the two pools. If we call the rates of incorporation from soluble and NPF-bound pools $k$ and $nk$, respectively, then the fraction of monomers incorporated from solution is simply $1/(n + 1)$. A lower bound for the value of the proportionality constant $n$ is simply the observed acceleration of network growth compared to the rate of elongation of an actin

filament in the same concentration of profilin–actin. In our experiments, actin networks grew from micropatterned WAVE surface 3.5-fold faster than individual filaments from the same concentration of soluble profilin–actin. From this comparison, we derive an upper bound on the fraction of actin incorporated from solution of 0.22. We therefore propose that membranes moved by the assembly of branched actin networks are active surfaces that collect actin monomers from solution and deliver them directly to the growing actin network.

The network polymerase activity of WASP-family proteins also adds an interesting complication to models aimed at describing the emergence of filopodia from dynamic lamellipodial actin networks. Both the *de novo* nucleation and convergent elongation models of filopodia formation (Welch & Mullins, 2002; Svitkina *et al*, 2003) require that filaments destined to form a filopodium must elongate faster than nearby lamellipodial filaments. Understanding how different actin polymerases compete with each other and how their self-association in the context of a lipid bilayer affects this competition will require additional work.

### The density of WASP-family proteins on cellular membranes is high enough to support network polymerase activity

To determine whether physiologically relevant densities of WASP-family proteins are sufficient to support network polymerase activity, we compared the density of WAVE1 molecules on our micropatterned surfaces to densities of WASP-family proteins estimated on various cellular membranes. In budding yeast, WASP densities are estimated to reach 25,000 molecules/$\mu m^2$ (Arasada & Pollard, 2011). Quantitative analysis of Nck-activated actin assembly (Ditlev *et al*, 2012) suggests that N-WASP densities reach 10,000 molecules/$\mu m^2$. However, considering the large footprint of the five-subunit WAVE regulatory complex on membranes (~19 × 8 nm; Chen *et al*, 2010), we estimate a geometric packing limit for this NPF of ~6,500 molecules/$\mu m^2$. The total cellular concentration of the WAVE complex is estimated to be 130 nM in neutrophil-like cells (Weiner *et al*, 2007). However, only 10% of the WAVE complex is estimated to translocate to the membrane during polarized cell migration (Weiner *et al*, 2007), with an effective leading edge membrane surface area of roughly 15 $\mu m^2$. Therefore, WAVE complexes occupy a surface area that is approximately 25% of the entire leading edge membrane. These assumptions lead us to estimate that WAVE complex membrane densities higher than 1,000 molecules/$\mu m^2$ are reached at sites of actin assembly. The densities of WAVE1 (PWCA) measured in our surface polymerization assays are approximately 2,000 molecules/$\mu m^2$, and about one order of magnitude higher (15,000 molecules/$\mu m^2$) on lipid membranes used for microsphere experiments, indicating that we are working at close to physiological conditions.

We currently do not know whether the polymerase function of NPFs is regulated by upstream signaling molecules in ways similar to their nucleation-promoting activity. The WH2 sequences of some NPFs (WAVE and WASP) appear to be tied up in intramolecular interactions in the autoinhibited state and become available for monomer binding only upon activation (Kim *et al*, 2000; Chen *et al*, 2010). The proline-rich domains, on the other hand, are either omitted from or not resolved in the available NPF structures, indicating that these might be sufficiently flexible to bind profilin–actin even in the absence of potential upstream activators. Importantly, the

polymerase mechanism established here relies on the local enrichment of NPF proteins at cellular membranes. Localization of most NPFs is controlled by the same molecules that activate them (Rho-type GTPases and signaling lipids). We therefore consider it unlikely that the majority of membrane-bound NPFs exists in an inactive state *in vivo*. On the other hand, several additional scaffold proteins, such as IRSp53 (Scita *et al*, 2008) or Nck (Ditlev *et al*, 2012; Banjade *et al*, 2015), are known to organize NPFs at sites of actin assembly through multivalent interactions. How clustering, and potentially phase separation (Li *et al*, 2012), of WASP-family proteins might influence their ability to promote actin elongation remains to be determined.

### Actin flows onto filament ends from both WH2 sequences and PRD-bound profilin–actin complexes

WASP-family WH2 domains bind monomeric actin while nearby proline-rich domains bind profilin–actin complexes. We find that actin from either source can incorporate into nearby barbed ends, and more interestingly, we discovered a functional collaboration between the two domains. Unoccupied WH2 sequences can dynamically engage with and transiently occupy the barbed ends of actin filaments, but they are displaced from filament ends by binding to monomeric actin. This phenomenon explains how WASP-family proteins can dynamically tether a growing actin network to the surface it pushes against (Co *et al*, 2007). Profilin plays an important role in setting the balance between tethering and polymerization by controlling access to the WH2 domain to monomeric actin. By competing for an overlapping binding site on actin, profilin decreases the monomer occupancy of the WH2 sequence (Chereau *et al*, 2005). This type of competition with profilin has been previously described for a number of WH2 containing proteins (Bosch *et al*, 2007; Heimsath & Higgs, 2012; Suarez *et al*, 2015). However, due to the ubiquitous use of minimal WCA constructs of nucleation-promoting factors that lack proline-rich sequences, the dual role of profilin in modulating filament tethering and polymerization activities through the WH2 domain has remained elusive. Mechanistically, this effect might be responsible for the observed inhibition of Arp2/3-dependent nucleation in response to excess profilin levels *in vivo* (Rotty *et al*, 2015; Suarez *et al*, 2015) and *in vitro* (Mullins *et al*, 1998), although other potential causes have been proposed as well (Pernier *et al*, 2016; Suarez & Kovar, 2016). Interestingly, profilin-mediated inhibition of Arp2/3 activity is not complete even at large profilin excess (Blanchoin *et al*, 2000; Rotty *et al*, 2015; Suarez *et al*, 2015). We propose that this might be explained by monomer transfer from the PRD to the WH2 domain, which allows NPFs to maintain some monomer-bound WH2 domains and sustain non-zero levels of Arp2/3 nucleation even in the presence of profilin. In conclusion, the presence of the proximal proline-rich domain alleviates—but does not completely compensate for—the inhibitory effect of profilin on WH2 occupancy.

### Monomer transfer and domain collaboration in filament-nucleating and filament-elongating proteins

Profilin–actin complexes probably constitute the majority of polymerizable monomers in most eukaryotic cell types (Kaiser *et al*, 1999). Free monomers, on the other hand, are unlikely to accumulate at substantial levels *in vivo* because of (i) their sub-micromolar

critical concentration and (ii) the high concentrations of monomer-binding proteins in the cytoplasm. It is therefore not surprising that nearly all actin elongation and nucleation factors identified thus far harbor PRD-like sequences to utilize the dominant monomer pool. Following the initial recruitment, however, profilin–actin needs to be funneled into diverse biochemical processes through transfer mechanisms that are best defined for processive polymerases. Both Ena/VASP and formin proteins shuttle profilin–actin *en bloc* from the PRD to the proximal barbed end they persistently bind to. Interestingly, profilin–actin transfer in Ena/VASP proteins involves binding to WH2-like globular actin-binding (GAB) domains that are one helical turn shorter than their WASP-family counterparts (Chereau *et al*, 2005; Chereau & Dominguez, 2006; Ferron *et al*, 2007). Shortening of the WH2 helix weakens the binding to monomeric actin, but eliminates the steric clash with profilin. In fact, the profilin–actin complex is the preferred substrate of the VASP GAB (Chereau & Dominguez, 2006). This small divergence in WH2 length is likely a necessary adaptation of VASP-family proteins that allows profilin–actin to be transferred as a unit, which enables rapid profilin–actin shuttling on the level of single proteins characteristic for these processive polymerases. In contrast, distributive polymerase activity of WASP proteins does not necessitate such a transfer mechanism, because PRD-associated profilin–actin is *sufficient* to promote elongation from membranes. Instead we consider monomer shuttling a prerequisite for the other two main functions of WASP proteins: nucleation and tethering. Intriguingly, WASP-family proteins likely need to separate monomers from profilin upon binding to their longer WH2 domains. Dissociation from profilin might be a requirement for WH2-bound monomers to participate in Arp2/3-mediated nucleation, as indicated by the failure of PRD sequences to substitute for WH2 function even in the presence of profilin. Also, competition with profilin for monomer binding allows NPFs to form a dynamic tether between the membrane and the dense actin network via their unoccupied WH2 domains.

Our work has revealed the diverse functions of WASP-family proteins in promoting branched network assembly (Fig 6). Each function originates from a distinct subset of modular domains within these proteins. Future comparative cell biology and biochemical analysis will uncover whether their activities can be uncoupled and thus be independently lost or maintained in evolution. For instance, do WASP-family proteins serve as polymerases in actin structures not generated by the Arp2/3 complex? Phylogenetic analysis, however, shows the domain architecture of WASP-like subfamilies is remarkably conserved. Also, WASP-family proteins and the Arp2/3 complex are both ancestral eukaryotic proteins which are only lost or maintained *together* (Veltman & Insall, 2010). We propose that these core proteins of the branched actin network motor function as a unit and therefore need to be understood at the network level.

## Materials and Methods

### Protein purification and labeling

#### WAVE1
The coding sequence of human WAVE1 was first codon-optimized for expression in *E. coli* (GeneArt, Invitrogen) and then used for the sub-cloning of fragments as indicated in Fig 3A.

Nucleation-promoting factor constructs used for surface immobilization (experiments on lipid-coated beads NPF-coated PEG surfaces) were fused to an N-terminal mCherry-tag (that acted as a fluorescent tag and spacer protein) harboring an N-terminal Lys-Cys-Lys- (KCK-) tag (for maleimide-PEG surface immobilization) followed by a His$_{10}$-tag (for purification and tethering to the lipid bilayer) and cloned into a modified pETM vector containing a TEV-cleavable z-tag. To prevent surface attachment via protein sites other than the N-terminal KCK-tag, endogenous cysteine residues of WAVE1 (Cys 296 and 407) were replaced with serine without affecting protein activity. Non-fluorescent versions of these mCherry-NPF fusions were generated by introducing a Y71S mutation in mCherry (darkCherry) to facilitate multicolor TIRF microscopy when direct visualization of the NPF was not desired. For the purification of mCherry or dark mCherry lacking NPF activity (mock proteins), we introduced a STOP codon between the Cherry and NPF moiety.

Constructs for experiments in solution were cloned into a modified pETM vector containing a TEV-cleavable His10-z-tag. For the detection of WH2 occupancy by FRET, a residue directly upstream of the WH2 domain (Thr490) was mutated to Cys.

All WAVE1 variants were expressed in *E. coli* (Star pRARE) for 16 h at 18°C and purified by IMAC over a HiTrap Chelating column, followed overnight TEV cleavage on ice, ion-exchange chromatography over MonoQ or MonoS columns and size exclusion over a Superdex 200 or 75 column. Constructs for FRET experiments were labeled at the introduced single Cys residue (Cys490) overnight on ice with a fivefold excess of Alexa488-maleimide before the final gel filtration steps. Proteins were snap-frozen in liquid nitrogen in storage buffer containing 20 mM HEPES [pH 7.5], 150 mM NaCl, 0.5 mM TCEP, 0.1 mM EDTA, 20% glycerol.

### N-WASP

The coding sequence of human N-WASP was first codon-optimized for expression in *E. coli* (GeneArt, Invitrogen) and then used for the sub-cloning of the PWWCA region (275–505).

For surface immobilization, the PWWCA region was fused to an N-terminal mCherry-tag (that acted as a fluorescent tag and spacer protein) harboring an N-terminal Lys-Cys-Lys- (KCK-) tag (for maleimide-PEG surface immobilization) followed by a His$_{10}$-tag (for purification) and cloned into a modified pETM vector containing a TEV-cleavable z-tag. To prevent surface attachment via protein sites other than the N-terminal KCK-tag, an endogenous cysteine residue (Cys 431) was replaced with serine without affecting protein activity. The proteins was expressed in *E. coli* (Star pRARE) for 16 h at 18°C and purified by IMAC over a HiTrap Chelating column, followed overnight TEV cleavage on ice, ion-exchange chromatography over MonoQ or MonoS columns and size exclusion over a Superdex 200 or 75 column. Protein was snap-frozen in liquid nitrogen in storage buffer containing 20 mM HEPES [pH 7.5], 150 mM NaCl, 0.5 mM TCEP, 0.1 mM EDTA, 20% glycerol.

### Actin

Native, cytoplasmic actin from *Acanthamoeba castellanii* was purified by ion-exchange chromatography and a cycle of polymerization–depolymerization as described previously (Hansen *et al*, 2013) and stored in filamentous form dialyzing against polymerization buffer (20 mM imidazole (pH = 7.0), 50 mM KCl, 1.5 mM MgCl$_2$, 1 mM EGTA, 0.5 mM ATP, 0.5 mM TCEP). 5 ml fractions of the filamentous pool were depolymerized at a time by dialyzing into G-Buffer (2 mM Tris–Cl (pH = 8.0), 0.1 mM CaCl$_2$, 0.2 mM ATP, 0.5 mM TCEP) for 1 week, followed by size exclusion using a HiLoad Superdex 200 (XK16-60) column. Actin was stored in monomeric form at 4°C for up to 2 months after gel filtration.

Actin was fluorescently labeled with Alexa488- or Cy5-maleimide at Cys 374 as previously described (Hansen *et al*, 2013). For labeling actin with Atto540Q-NHS, the profilin–actin complex was formed in G-Buffer with a 1.5-fold excess of profilin. The complex was isolated by size exclusion using a Superdex 75 column in labeling buffer (2 mM HEPES (pH = 8.0), 0.1 mM CaCl$_2$, 0.2 mM ATP, 0.5 mM TCEP), concentrated and labeled at reactive lysine residues by incubating with a 10-fold excess of the NHS-dye for 1 h on ice. After quenching with Tris–Cl (2 mM, pH = 8.0), actin was polymerized by addition of 10× polymerization buffer and a small quantity (1% of total actin) of freshly sheared filaments that acted as seeds for filament elongation. After polymerization for 1 h at room temperature, filaments were pelleted by ultracentrifugation (20 min at 278,000 × *g* in a TLA100.2 rotor) and then depolymerized in G-Buffer for 1 week in the dark. Depolymerized, Atto540Q-actin was then gel-filtered over a Superdex 75 column and stored on ice.

### Arp2/3 complex

The native, bovine Arp2/3 complex was purified from calf thymus glands (PelFreez) by a series of ammonium sulfate precipitation and ion-exchange chromatography (DEAE, Source Q and Source S) steps followed by size exclusion chromatography (Superdex 200) as described previously (Doolittle *et al*, 2013). Proteins were snap-frozen in liquid nitrogen in storage buffer (5 mM HEPES (pH = 7.5), 50 mM NaCl, 0.5 mM MgCl$_2$, 0.5 mM TCEP, 0.5 mM EGTA, 0.1 mM ATP, 20% glycerol).

### Capping protein

To generate wt CP, the α1 and β2 isoforms of murine heterodimeric capping protein were cloned into pETM20 and pETM33, respectively. Proteins were co-expressed in *E. coli* (Rosetta) for 16 h at 18°C and purified by IMAC over a 5 ml HiTrap Chelatin column followed by overnight TEV/Prescission cleavage of the N-terminal His-tags on ice. After desalting over a HiLoad Desalting column, uncleaved protein and free tags were removed by recirculation over the IMAC column. The flow through was subjected to ion-exchange chromatography over a Mono Q column and gel filtration over a Superdex 200 column. Proteins were snap-frozen in liquid nitrogen in storage buffer (10 mM Tris–Cl (pH = 7.5), 50 mM NaCl, 0.5 mM TCEP, 20% glycerol).

### Profilin

Human profilin 1 or profilin 2 was expressed in *E. coli* (BL21) for 6 h at 30°C and purified by ammonium sulfate precipitation, followed by ion-exchange (DEAE) and hydroxylapatite (HA) chromatography steps, followed by size exclusion chromatography (Superdex 75). Proteins were snap-frozen in liquid nitrogen in storage buffer containing 10 mM Tris [pH 8.0], 50 mM KCl, 1 mM EDTA, 0.5 mM TCEP, 20% glycerol.

### Utrophin actin-binding domain

The actin-binding domain (UTRN$_N$, AA 1–261) of human utrophin was cloned into a modified pETM vector containing an N-terminal

TEV-cleavable His10-z-tag and a C-terminal Lys-Cys-Lys-(KCK)-tag for maleimide labeling. Protein was expressed in *E. coli* (BL21) for 16 h at 18°C and purified by IMAC over a 5 ml HiTrap Chelatin column followed by overnight TEV cleavage of the N-terminal His10-z-tag on ice. After desalting over a HiLoad Desalting column, uncleaved protein and free tags were removed by recirculation over the IMAC column. The flow through was subjected to labeling with a fivefold excess of Alexa488-maleimide overnight, followed by gel filtration over a Superdex 200 column. Proteins were snap-frozen in liquid nitrogen in storage buffer containing 10 mM Tris–Cl [pH 7.5], 50 mM NaCl, 0.5 mM TCEP, 20% glycerol.

### Coverslip functionalization, photolithography, and protein immobilization

Glass coverslips (22 × 22 mm, #1.5, high precision, Zeiss) were functionalized (Bieling *et al*, 2010) and patterned (Fourniol *et al*, 2014) essentially as described previously. Briefly, surfaces were rigorously cleaned by consecutive incubation in 3 M NaOH and Piranha solution (3:2 concentrated sulfuric acid to 30% hydrogen peroxide) followed by silanization with (3-glycidyloxypropyl) trimethoxysilane.

For pure PEG-biotin functionalization, silanized surfaces were passivated by reacting for 4 h at 75°C with a 150 mg/ml mixture of hydroxyl-PEG$_{3000\ Da}$-NH$_2$ (95mol%) and CH$_3$O-biotin-PEG$_{3000\ Da}$-NH$_2$ (5 mol%) in dry acetone. The PEG-biotin coverslips were then washed with copious amounts of Milli-Q water before spin drying and storage in a dust-free container at room temperature.

For combined biotin- and WAVE1-PEG functionalization, silanized surfaces were passivated by reacting with a 150 mg/ml mixture of hydroxyl-PEG$_{3000\ Da}$-NH$_2$ (80%), NH$_2$-PEG$_{3000\ Da}$-NH$_2$ (5%), and CH$_3$O-biotin-PEG$_{3000\ Da}$-NH$_2$ (15%) in anhydrous acetone. Subsequently, exposed amino groups were reacted with a heterobifunctional crosslinker (BMPS) to create hybrid PEG-maleimide/biotin-coated coverslips. For experiments on micropatterned surfaces, PEG-maleimide/biotin-coated coverslips which were subjected to UV microlithography using a chrome-on-quartz photomasks as described (Fourniol *et al*, 2014). Micropatterned or evenly coated PEG-maleimide/biotin coverslips were then loosely attached to flow chambers constructed of PLL-PEG passivated microscopy counter slides and thin PDMS stripes (flow cell volume = 40 μl). For the immobilization of WAVE1 on PEG-maleimide/biotin coverslips, protein aliquots of WAVE1 proteins were rapidly thawed and pre-reduced with 1mM beta-mercaptoethanol for 30 min on ice and then desalted twice into immobilization buffer (20 mM HEPES (pH = 7.5), 300 mM NaCl, 0.5 mM EDTA). Protein concentration was determined by absorbance (280 nm), and WAVE1 protein mix was prepared by diluting desalted proteins to 10 μM total in immobilization buffer, followed by direct incubation for 25 min at room temperature with the freshly prepared PEG-maleimide/biotin coverslips in the flow cell contained in a humidified chamber. After protein immobilization, flow cells were washed with 6 flow cell volumes wash buffer (20 mM HEPES (pH = 7.5), 300 mM NaCl, 0.5 mM EDTA, 5 mM beta-mercaptoethanol), incubated for 3 min to quench residual maleimide groups, washed with 6 flow cell volumes storage buffer (20 mM HEPES (pH = 7.5), 300 mM NaCl, 0.5 mM EDTA, 2 mM TCEP) and stored at 4°C in a humid container for up to 5 days.

The average density of his$_{10}$-Cherry-WAVE1 PWCA molecules on the PEGylated surface was determined by fluorescence fluctuation spectroscopy. Briefly, we performed a statistical analysis of the distribution of photon counts per step obtained from a scanning a small region of the sample with a confocal microscope. Regions were scanned with a grid of 64 × 64 steps. The average number of counts per step (<k>) and the variance in the number of counts per step <Dk$^2$> were calculated. The average number of molecules in the laser spot <N> is then given by (Delon *et al*, 2006):

$$<N> = 1/2 <k>^2 / (<Dk^2> - <k>).$$

The molecular concentration (C) is given by <N> divided by the area of the laser spot (Delon *et al*, 2006):

$$C = 2 <N> / (pw_0^2)$$

where w$_0$ is the 1/e$^2$ radius of the focused laser beam (Chen *et al*, 1999). The laser beam radius of 200 nm was determined by scanning sub-diffraction sized fluorescent beads. Scans of five regions of the sample were used to calculate the average molecular surface density, which was 1200 ± 320 molecules/μm$^2$. Because only 67% of all mCherry fluorophores were active after purification from *E. coli*, we divided the surface density of active fluorophores by this fraction to yield the total surface density, which was 1,850 ± 500 molecules/μm$^2$.

### Fluorescence microscopy

All TIRF microscopy data were collected on a Nikon Eclipse Ti microscope (Nikon Instruments) microscope equipped with a total internal reflection fluorescence (TIRF) illuminator (Nikon), a TIRF objective (100×, numerical aperture 1.49), a cooled charge-coupled device camera (iXon, Andor), and a hardware-based Nikon Perfect focus system. Fluorescence excitation was accomplished by three diode-pumped solid-state laser lines (488, 561, and 644 nm), which were controlled using an acousto-optical filter and coupled into a single fiber-optic light guide (custom laser launch, Spectral Applied Research). Micro-Manager (Edelstein *et al*, 2010) was used to control the shutters, acousto-optical filter, dichroic mirrors, and camera. Laser intensity and exposure were minimized to avoid photo-bleaching. For multicolor fluorescence measurements of single actin filaments, images (between 20 and 150 ms exposure time) were taken at custom intervals of time (1–15 s) to avoid overexposure and bleaching.

Confocal imaging was performed on an Observer.Z1 (Zeiss) microscope equipped with a confocal scanner (CSU-X1; Yokogawa Electric Corporation), a 63× objective (Plan-Apochromat 63× 1.4, Zeiss) and a cooled charge-coupled device camera (CascadeII, Photometrics). Fluorescence excitation was accomplished by three diode-pumped, solid-state laser lines (401, 488, 561, and 644 nm), which were controlled using an acousto-optical filter and coupled into a single fiber-optic light guide (custom laser launch, Solamere Technology Group). Micro-Manager (Edelstein *et al*, 2010) was used to control the shutters, acousto-optical filter, dichroic mirrors, and camera. Laser intensity and exposure were minimized to avoid photo-bleaching. Multicolor *z*-stacks (100 ms exposure time at 0.2–1 mM step sizes) were acquired at 1-min time intervals.

Wide-field epifluorescence microscope images of dendritic actin networks assembled on lipid-coated beads were visualized using an inverted Nikon Eclipse TE-2000E microscope with a 60× Nikon (NA 1.49) water immersion objective. Images were acquired on a 2048 × 2048 pixel CoolSnap Photometrics camera using Micro-manager 1.4 software (Edelstein *et al*, 2010).

### Single-filament assays on passivated coverslips

Single-filament assays were essentially performed as described previously (Hansen *et al*, 2013). In brief, phalloidin-stabilized actin filament seeds were flowed into the imaging chamber and captured by biotinylated heavy meromyosin (biotin-HMM), which was immobilized to streptavidin-coated biotin-PEG surfaces. Unattached actin filaments were washed out of the imaging chamber. Actin polymerization reactions were initiated by combining 1 µl of 10× ME (20 mM EGTA, 5 mM $MgCl_2$) with 9 µl of 4.44x-stock solution of monomeric actin (0–20% Cy5 labeled) for 2 min at room temperature. The Mg-ATP-actin was then combined with the TIRF buffer mix containing other additives (Alexa488-UTRN$_N$, profilin etc., as indicated in the figure legends and main text) to reach the final actin concentration. Final buffer composition was 20 mM HEPES (pH = 7.0), 100 mM KCl, 20 mM beta-mercaptoethanol, 1.5 mM $MgCl_2$, 1 mM EGTA, 1 mM ATP, 0.5 mg/ml beta-casein, 0.2% methylcellulose (cP400, M0262; Sigma-Aldrich), 40 mM glucose, 125 µg/ml glucose oxidase (Bio-phoretics), 20 µg/ml catalase (Sigma-Aldrich), and 2 mM Trolox (Sigma). The glucose oxidase and catalase solutions were made fresh from dry reagents, ultracentrifuged to remove debris (TLA120.1, 278,587 × *g*, 20 min) and used within 3 days. To initiate filament elongation, the mixture of Mg-ATP actin diluted in TIRF imaging buffer was flowed into the imaging chamber, transferred to the microscope and filaments were visualized immediately.

### Single-filament assays on WAVE1-immobilized, passivated coverslips

Flow cells of micropatterned, WAVE1-coated coverslips (see coverslip functionalization above) were washed with twice with 250 µl of ultra-pure Milli-Q water and disassembled by removal of the coverslips. Excess water was removed by a brief (5 s) spin on a spin coater. Drying did not affect NPF activity if the coverslip was not kept in air for > 30 min. The coverslip was fixated on a custom-made flow cell, incubated with streptavidin followed by biotin-HMM and transferred to the microscope stage. Phalloidin-stabilized actin filament seeds were flowed into the imaging chamber and captured by the surface-bound HMM. Unattached actin filaments were washed out of the imaging chamber. Actin polymerization reactions were initiated by combining 1 µl of 10× ME (2 mM EGTA, 0.5 mM $MgCl_2$) with 9 µl of 4.44x- stock solution of monomeric actin (0–20% Cy5 labeled) for 2 min at room temperature. The Mg-ATP-actin was then combined with the TIRF buffer mix containing other additives (Alexa488-UTRN$_N$, profilin etc.) as indicated in the figure legends and main text to reach the final actin concentration. To initiate filament elongation, the mixture of Mg-ATP actin diluted in TIRF imaging buffer was flowed into the imaging chamber, transferred to the microscope and filaments were visualized immediately. Final buffer composition was as described in single-filament assays on passivated coverslips (see above).

### Dendritic network assembly on micropatterned coverslips

Flow cells of micropatterned, NPF-coated coverslips were washed with twice with 250 µl of ultra-pure Milli-Q water and disassembled by removal of the coverslips. Excess water was removed by a brief (5 s) spin on a spin coater. Drying did not affect NPF activity if the coverslip was not kept in air for > 30 min. The coverslip was fixated on a custom-made flow cell and transferred to the microscope stage. The NPF pattern (visualized by the Cherry-NPF fluorescence) was then positioned in the center of the field of view. Actin network growth was initiated by addition of network proteins in assembly buffer (final concentration: 5 µM actin (1% Cy5-Actin), 5 µM profilin, 50 nM Arp2/3, 200 nM CP). Synchronously, confocal time-lapse imaging was initiated. Final buffer composition was as described in single-filament assays on passivated coverslips (see above).

### Preparation of small unilamellar vesicles

The following lipids were used to make small unilamellar vesicles: 18:1 (Δ9-Cis) PC (DOPC) 1,2-dioleoyl-sn-glycero-3-phosphocholine (Avanti #850375C) and 18:1 DGS-NTA(Ni) [1,2-dioleoyl-sn-glycero-3-[(N-(5-amino-1-carboxypentyl)iminodiacetic acid)succinyl] (nickel salt) (Avanti #790404C). Lipids received were received a single-use ampules dissolved in chloroform. Liposomes were prepared by combining 1–2 ml of chloroform and 2 µmoles total lipids in a glass round bottom flasks, before drying with a rotovap. Follow rotovap mediated evaporations of chloroform, round bottom flasks containing lipid films were dried under house nitrogen gas for 30 min or overnight under vacuum. Lipid films were then resuspended with phosphate-buffered saline [pH 7.2] to concentration of 1 mM and extruded through a 30 nm diameter polycarbonate filter (Avanti) with a total of 11 passes. Vesicles were used the same day.

### Generation of Lipid-coated glass microspheres

As previously described (Hansen & Mullins, 2015), glass microspheres were washed with concentrated nitric acid by diluting a 10% wt/vol slurry of 2.34 µm silica beads (Bangs Laboratories, Cat# SS05N) to 1% wt/vol in a borosilicate glass vial and incubated for at least 3 h at room temperature. Nitric acid-washed glass beads were then pelleted by centrifugation for 5 min at 500 × *g*. Microspheres were washed four times in a glass vial with Milli-Q water. The process of centrifugation and washing was repeated four times. Finally, beads were resuspended in water to final concentration of 10% wt/vol. Lipid bilayers was assembled on acid-washed glass beads by combining 20 µl of 10% bead slurry (vortex/sonicate before aliquoting) with 105 µl 20 mM HEPES [pH 7], 150 mM NaCl (or PBS [pH 7.2]) in an Eppendorf tube. Diluted glass beads were vortexed and then bath sonicated for 5 min. Monodisperse glass beads were then combined with 25 µl of 1 mM SUVs, vortexed briefly, and then rotated at room temperature for 30 min. After assembling the lipid bilayer, 750 µl of Milli-Q water was added to each tube and beads were micro-centrifuged for 2 min at 200 × *g*. The supernatant was aspirated off the beads and the 750 µl Milli-Q water wash, followed by micro-centrifuged was repeated four times. After the final spin, the supernatant was aspirated leaving ~50 µl of water/beads. Beads were then resuspended by vortexing, and 150 µl of buffer containing 20 mM HEPES [pH 7], 200 mM KCl was added. The final bead slurry was ~1% (wt/vol) and contained a final KCl concentration of 150 mM. To charge lipid-coated glass

beads with protein, we combined 5 μl of 1% bead slurry with 45 μl of 111 nM $his_{10}$ tagged protein (i.e., $his_{10}$-Cherry-WAVE PWCA). Proteins were diluted into buffer containing 20 mM HEPES [pH 7], 150 mM KCl, 100 μg/ml BSA, 0.5 mM TCEP.

The average density of $his_{10}$-Cherry-WAVE PWCA bound to lipid-coated glass beads via the Ni-NTA lipid interaction was calculated using a fluorimeter. First, the total number of LCBs per ml was calculated using a hemocytometer to count the beads on Nikon Eclipse TE300 light microscope (20× Apo Fluor Nikon objective, NA 0.45). A defined number of beads (~1% final bead slurry or $2.7 \times 10^6$ total beads) were mixed with a final solution concentration of 100 nM $his_{10}$-Cherry-WAVE PWCA. Following a 15-min incubation at 23°C, the sample was centrifuged at $21,000 \times g$ for 2 min to pellet the LCB. The supernatant was removed and analyzed using a Varian Cary Eclipse fluorimeter to detect mCherry fluorescence emission at 610 nm. In parallel, a $his_{10}$-Cherry-WAVE PWCA fluorescence standard curve (0–150 nM) was generated using purified protein. Typically, ~15% of the total $his_{10}$-Cherry-WAVE PWCA is depleted from solution in the presence of LCBs (lipid composition: 98% DOPC and 2% Ni-NTA). Using wide-field fluorescence microscopy, we verified that LCBs were monodispersed and uniform in mCherry fluorescence following the 15-min incubation with $his_{10}$-Cherry-WAVE PWCA. The average density of $his_{10}$-Cherry-WAVE PWCA was calculated assuming all beads are 2.34 μm in diameter. Assuming the lipid bilayer increases the bead diameter by 0.005 μm, then the total surface area of each LCB equals 17.3 μm$^2$. Based on the (i) total number of beads, (ii) total number of $his_{10}$-Cherry-WAVE PWCA molecules depleted from solution, and (iii) average bead surface area, we estimate an average density of ~15,000 $his_{10}$-Cherry-WAVE PWCA per μm$^2$. This density corresponds to 55% of the Ni-NTA lipid binding sites being occupied on each LCB.

### Dendritic network assembly on lipid-coated beads

As previously described by (Hansen & Mullins, 2015), dendritic network assembly was initiated by mixing WAVE1-coated LCBs with soluble proteins reactants (Arp2/3, CP, profilin–actin). Conditions were 5 μM actin, 5 μM profilin I, 50 nM CP, 200 nM Arp2/3. Buffer was 20 mM HEPES pH 7, 100 mM KCl, 1 mM MgCl$_2$, 1 mM EGTA, 1 mM ATP, 0.2% methylcellulose (cP 400), 2.5 mg/ml BSA, 20 mM beta-mercaptoethanol. Following 5–10 min of actin network assembly, reactions were typically quenched and flowed into a microscope imaging chamber, which was then sealed with VALAP. We quenched actin assembly and disassembly on lipid-coated glass beads by combining equal volumes of the bead motility reaction and 25 μM Latrunculin B-phalloidin (1:1, therefore 5 molar excess relative to concentration of actin) diluted in buffer containing 20 mM HEPES [pH 7], 100 mM KCl, 100 μg/ml BSA, 0.5 mM TCEP. Imaging chamber for the bead motility assay was assembled with glass silanized with a 2% solution of diethyldichlorosilane (Gelest, Cat# SID 3402.0) in isopropanol (pH 4.5 with acetic acid) (Akin & Mullins, 2008).

### Analytical ultracentrifugation

All experiments used human profilin I (PfnI) and mouse profilin 2a (Pfn2a) with variants of human WAVE1 [277–488]. To incorporate a fluorescent label at a specific position, two endogenous cysteines (C296 and C407) WAVE1 [277–488] were mutated to serines and an

N-terminal KCK motif or a single cysteine was introduced. We refer to this construct as WAVE1 (P). We identified six putative profilin-binding sites (A–F, starting from the N-terminus) in WAVE1 and replaced endogenous prolines with GS repeats such that all stretches of three or more consecutive prolines were eliminated. We refer to this construct as WAVE1 PRD [Null]. WAVE PRD[A] through [F] retain only the named wild-type sequence with the remaining sites mutated as in the Null construct. The WAVE constructs used in sedimentation equilibrium experiments were labeled with Texas Red maleimide (for the KCK variants) or Alexa 546 maleimide (for the C variants) as indicated.

Sedimentation equilibrium (SE) experiments were performed using six-well Teflon chambers with quartz windows and placed in a four-channel Ti-60 rotor. We monitored solute distributions with the absorbance optics (556 nm for Texas Red and 521 nm for Alexa 546) of a Beckman XL-I analytical ultracentrifuge. The equilibrium solute distributions were measured for three speeds (10 K, 14 K, and 20 K rpm for profilin:WAVE1(P) SE; 7 K, 10 K, and 14 K for actin: WAVE1(P) SE). In all cases, the ultracentrifuge was operated at 20°C. To minimize the effect of protein degradation, a modified version of the overspeed protocol by was implemented. Protein stability was confirmed by comparing pre- and post-run samples with SDS–PAGE. The measured molecular weight for all WAVE1 constructs was within 3% of the expected value. The base buffer composition for all SE runs was 25 mM HEPES (pH 7.0), 50 mM KCl, 1 mM MgCl$_2$, 1 mM EGTA, 0.5 mM TCEP. Buffer density and protein partial specific volumes were calculated using Sednterp. Global fitting of three equilibrium traces from all speeds (i.e., 7, 10, 14 K rpm) for each condition was performed using open source NIH Sedphat and Sedphat software (Peter Schuck, NIH). We used baseline absorbance, radius-independent noise (RI), time-independent noise (TI), and cell bottom position as fitting parameters with mass conservation constraints. Protein concentrations and extinction coefficients were fixed to measured and known values.

### Fluorescence polarization and FRET assays

For equilibrium actin-binding assays, Alexa488-labeled WAVE1 (7.5 nM) was mixed with Latrunculin B-stabilized unlabeled or Alexa540Q-labled actin at the indicated concentrations on ice, transferred to room temperature and incubated for 5 min. Steady state anisotropy or FRET data were then collected with a K2 Multifrequency Fluorometer. The Alexa488 donor was excited at 487 nm, and donor emission was detected using a 515/35 nm bandpass filter (ET series, Chroma). For the determination of anisotropy, we excited the fluorophore with plane polarized light and measured emission at polarizations both parallel and perpendicular to the excitation source. Under the conditions used in our study, anisotropy is a measure of the rotational mobility of the fluorescently labeled protein. For the measurement of FRET efficiency, we determined the drop in donor (Alexa488) intensity caused by energy transfer to the non-fluorescent quencher (Atto540Q). For experiments with non-fluorescent competing ligands, we premixed Alexa488-WAVE1:Atto540Q-actin on ice at concentrations of 7.5 and 175 nM, respectively. We then added competing ligands at indicated concentrations on ice and incubated the mixture for 5 min at room temperature before proceeding to measurements. Final buffer composition for all assays was 20 mM HEPES (pH = 7.0), 100 mM

KCl, 20 mM beta-mercaptoethanol, 1.5 mM $MgCl_2$, 1 mM EGTA, 1 mM ATP, 0.5 mg/ml beta-casein.

### Data analysis

Barbed end growth velocities for single actin filaments were determined from time-lapse TIRF imaging using the Kymograph toolbox (EMBL Heidelberg) in ImageJ. At least 50 filaments from two independent experiments were analyzed per condition.

Network growth velocities on micropatterned WAVE surfaces were analyzed from axial reconstructions from spinning-disk confocal time-lapse imaging using manual line scans in ImageJ. At least 50 networks were analyzed per condition.

The rate of dendritic actin network assembly on lipid-coated glass beads was quantified using ImageJ. Actin comet tail lengths were measured manual using the line draw tool. Pixel lengths were then converted to microns or microns/minute tail growth.

Fluorescence anisotropy and FRET data were analyzed as described (Zalevsky *et al*, 2001). Briefly, equilibrium dissociation constants were determined by fitting the binding data to a quadratic binding model. In experiments where non-fluorescent ligands compete with fluorescent Alexa488-WAVE1, we determined equilibrium binding constants by fitting a one-site competition model.

**Expanded View** for this article is available online.

### Acknowledgements

We thank members of the Fletcher and Mullins labs for discussions. This work was supported by NIH R01 GM074751 (D.A.F.), NIH R01 GM061010 (R.D.M), HHMI (R.D.M.). HFSP LT-000843/2010 (P.B), EMBO ALTF 854-2009 (P.B). T.-D.L. was supported by the Taiwan National Science Council.

### Author contributions

PB and SDH prepared all reagents, designed and performed experiments, analyzed data and wrote the manuscript. OA performed and analyzed the analytical ultracentrifugation experiments and contributed important early observations that initiated the project. T-DL assisted with network experiments on NPF micropatterns. CCH analyzed surface densities of by NPF micropatterns by fluorescence fluctuation spectroscopy. DAF and RDM supervised the project and wrote the manuscript.

### Conflict of interest

The authors declare that they have no conflict of interest.

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
