## [Review Process File · The EMBO Journal]

Manuscript EMBO-2017-97039

WH2 and proline-rich domains of WASP-family proteins collaborate to accelerate actin filament elongation

Peter Bieling, Scott D. Hansen, Orkun Akin, Tai-De Li, Carl C. Hayden, Daniel A. Fletcher and R. Dyche Mullins

Corresponding authors: Peter Bieling, Daniel A. Fletcher and R. Dyche Mullins, University of California

Review timeline:

Submission date:	29 March 2017
Editorial Decision:	05 May 2017
Revision received:	08 August 2017
Editorial Decision:	16 September 2017
Revision received:	19 September 2017
Accepted:	20 September 2017

Editor: Ieva Gailite

Transaction Report:

1st Editorial Decision

05 May 2017

Thank you for submitting your manuscript for consideration by the EMBO Journal. We have now received three referee reports on your manuscript, which are included below for your information. As you can see from the comments, the referees appreciate the proposed mechanism of WASP-mediated actin polymerisation. Given the reviewers' positive recommendations, I would like to invite you to submit a revised version of the manuscript, addressing the comments of all referees. I should add that it is The EMBO Journal policy to allow only a single major round of revision and that it is therefore important to resolve the main concerns at this stage.

When preparing your letter of response to the reviewers' comments, please bear in mind that this will form part of the Review Process File, and will therefore be available online to the community. For more details on our Transparent Editorial Process, please visit our website: http://emboj.embopress.org/about#Transparent_Process

Thank you for the opportunity to consider your work for publication. I look forward to your revision.

REFEREE REPORTS

Referee #1:

This is a beautiful work, which is both technically elegant and makes a major functional discovery; it conclusively demonstrates that, in addition to activating the Arp2/3 complex, WASP-family NPFs can accelerate barbed end filament elongation and link dendritic actin networks to membranes when clustered on surfaces. While the work is all in vitro, it replicates conditions that are physiologically realistic, as NPFs are often clustered on membranes. I enthusiastically recommend publication of this work in EMBO J, and would additionally suggest that the paper be highlighted.

My recommendations below are for the most part minor, and are mostly intended to improve the presentation.

1. My most significant concern is that the generalizing conclusion "WASP-family proteins can act as network polymerases and tethering factors" lacks experimental support. The paper only deals with one member of the WASP-family, WAVE/Scar. Therefore, at least one more NPF should be analyzed - preferably one with multiple WH2 domains. Alternatively, the conclusions should be less general.
2. Add page numbers.
3. Introduction: "All of these nucleation promoting factors share a set of conserved sequences, collectively known as a WCA motif, that mediate interactions with profilin, actin, and the Arp2/3 complex" The conserved core of all NPFs is PWCA, and this is something that should probably be stated right from the start, since it is central to this study (instead of talking about PRDs in a separate paragraph, which can be harder to understand for the non-initiated).
4. Introduction: "Efficient filament formation requires simultaneous association of an Arp2/3 complex with two actin monomer-loaded WCA sequences (Gaucher et al., 2012; Padrick et al., 2008, 2011; Ti et al., 2011)." One of these references is incorrect and two are missing. Note that the two NPF model was actually disputed by Gaucher et al., 2012, and was subsequently conclusively demonstrated by Boczkowska Nat Commun. 2014 and Helgeson JBC 2014.
5. It would be preferable to add a diagram as part A of Figure 3 detailing the specific nature of all the WAVE1 constructs used in the experiments. This is now given in Supplementary Figure 2 (a simpler version can do), which is not even cited in the main text.
6. Actually, several of the Supplemental Figures are not cited in the main text, and both 'Supplemental' and 'Supplementary' are used (not sure which is right).
7. Concerning the experiments in Figure 3 - consider disabling (as opposed to deleting) internal domains. Internal deletions change more than just the presence/absence of a domain, which can have unintended effects and obscure the conclusions.
8. In the diagrams of Figures 4 and 5, both profilin and the PRD regions are colored yellow. Because they interact with each other, using two different colors would be preferable.
9. Discussion: In the section entitled "The density of WASP-family proteins" it would be important to note that WASP-family proteins are subjected to activation, which could dramatically impact the actual density of these molecules that can function as polymerases. Moreover, the proteins that activate NPFs also take part of to the available membrane surface in such clusters. Finally, NPFs are often recruited to membranes through interactions with BAR domain proteins, most notably IRSp53, which are dimeric and form themselves membrane coats, with could profoundly impact the NPF polymerase mechanism proposed here.
10. Discussion: Concerning the parallels between WASP-family proteins and Ena/VASP and the transfer of profilin-actin from PRDs to W sequences the authors could also check Chereau J. Struct Biol 2006 for insights.
11. Discussion: This work demonstrates that the WH2 domain is a potent polymerase of free actin, and the proline-rich domain is a potent polymerase of profilin-actin. Therefore, it would be extremely helpful if the authors could expand on the roles of the WH2 vs. PRD regions within the context of known actin-profilin vs. actin monomer concentrations in cells.
12. It is also important to note that different NPFs contain different numbers of WH2 domains. How would this impact the WH2 vs. PRD interplay described above?

13. Discussion: It would be helpful if the authors could discuss the reasons for the inhibitory modulation of actin assembly by NPFs proposed by Pernier 2016 and Sweeney 2015 and comment on the differences/similarities with the stimulatory modulation proposed by Khanduja and Kuhn, 2014.

Referee #2:

This manuscript by Bieling et al. shows that proteins of the WASP family, which are mostly known as activators of the Arp2/3 complex responsible for the branched nucleation of actin filaments, are also able to accelerate barbed end elongation from actin or profilin-actin. The results are solid and confer a new status to this class of proteins. They should thus be of interest to a broad readership. In particular, the observations carried out with a pattern of WAVE1 proteins are quite spectacular and convincing.

However, I have some minor concerns about the work and would like to see the following points clarified before fully recommending publication.

1. The surface density of WAVE1 constructs seems to be a key point, and some information should be added. How the value of 1000 molec/micron² was determined should be briefly summarized. Have experiments been performed with different densities? The authors seem to say that a low amount of WAVE1 bound to the passivated region of the coverslip could account for the observed enhancement of the elongation rate with respect to the "negative control" (Fig. 2E). Can the amount of WAVE1 bound outside the pattern be estimated (using Cy5-actin fluorescence, maybe)? It seems that this surface density would be very low, yet it would be enough to cause a 2-fold increase in elongation rate. This should be discussed.

For measurements on dendritic networks, were the densities on microspheres and patterns the same? It seems to me that differences in surface densities could be as important as the geometrical factors invoked by the authors. In particular, the mechanical stress induced by the convex surface of the lipid-coated microspheres is certainly reduced by the fact that the surface is fluid, thereby allowing the NPFs to be mostly on one side of the microsphere, facing the actin "comet tail".

2. The claim that "in the absence of actin monomers the WAVE1 WH2 domain caps and tethers filament barbed ends" (subsection title) is not supported by the data. The authors show that WH2 domains slow down filament elongation from profilin-actin but, as they mention in the end of this subsection, this could result from a dynamic competition where WH2 would transiently occupy the barbed ends (similar to the "transient and opportunistic" interaction that is proposed earlier in the manuscript). Capping and tethering, which refer to a long-lasting bond between WH2 domains and barbed ends, cannot be excluded but are not demonstrated. In fact, they appear to be unlikely, since the rapid elongation promoted by WH2 domains in the presence of G-actin would rather require a rapid detachment of WH2 from barbed ends after a subunit has been added.

3. It is unclear to me what model the authors have in mind, exactly, to globally account for their observations. A sketch, summarizing their model for WAVE-induced accelerated elongation, would help.

What are the molecular details that account for the rapid elongation of filaments when they enter the WAVE1-patterned regions of the coverslip? Is the local increase in actin monomer concentration simply due to the monomers that are bound to the surface via WH2 domains, and which bind to growing barbed ends? (by the way, would this imply that WH2 rapidly detaches from its monomer after it has bound a barbed end?) Or are these monomers in rapid exchange between the solution and the surface, thus maintaining a local increase of monomer concentration in a small volume near the surface? Additional information on these kinetics could be obtained from FRAP measurements on the surface bound Cy5-actin (or by washing out the sample with buffer).

Of course, similar questions can be formulated for profilin-actin and proline-rich domains.

Is a processive interaction with the barbed end at all possible, similar to what Richard Dickinson

proposes in his models?

4. The strategy for filament tethering is not clearly presented. The main text refers to biotin-streptavidin linkages, but the methods specify that HMM (which is not defined - I suppose it is heavy meromyosin) was used to anchor the filaments. Things could be made clearer by explicitly showing the filament anchoring strategy in the sketches introducing the different experiments (Figs 1A, 2A, 3A and 3E). Are the anchoring point densities the same inside and outside the WAVE1 patterns?

Other points:

a. In the caption of Figure 4C, I do not understand the last sentence "... were referenced against the PWCA all cases, g."

b. I find the "shadow effect" added in various 2D cartoons (e.g., around coverslip in Fig 1E, around proteins in Figs 3A, 3C, 4A, 4D, 5B-D) useless and even counter-productive in terms of clarity.

c. I suppose that the affinities (dissociation constants) of actin for WH2 domains are determined by the fits in Fig 5 (while profilin's affinity for the binding sites in the PRD are determined by analytical ultracentrifugation). This should be said more clearly in the main text and in the caption of Fig. 5.

Referee #3:

This is an interesting paper that demonstrates a new function of WASP family proteins in regulating actin filament dynamics. In brief, the authors show that actin filaments growing on surfaces containing high densities of WASP-family proteins elongate much faster than those growing on other environments. They demonstrate that this effect is likely due to the ability of WH2-bound or PRD-bound actin or profilin-actin complex, respectively, to be fed onto the growing barbed end. They also show that PWCA constructs of WAVE1 can bind profilin-actin complexes in a manner that involves simultaneous contacts to both the PRD and WH2 elements. In general this is a well-performed, fairly complete and interesting study. I have several comments about the authors' overall presentation and interpretation of some of the data that should be addressed before publication.

My main criticism is that throughout the paper the authors emphasize that dendritic actin networks grow faster than individual filaments. This occurs in the abstract, the results, the figure legends and the discussion. But unless I am missing something important, this is not what the data show. The data show that filaments growing on dense WAVE clusters grow faster than those growing on other substrates. The difference in filament architecture in the initial experiments--network vs single filament--is incidental to the actual mechanism. So while it may be historically true that a comparison between individual filaments growing on a PEG-coated coverslip and a dendritic network growing on a WCA-coated bead initiated the project, it is misleading to readers to introduce the work as comparing single filaments to dendritic networks. This comes up over and over (e.g. abstract sentence 3, legend to figure 1, start of the results, start of the discussion, etc.), and really needs to be corrected. Even the name given to the phenomenon--"network polymerase" activity--is misleading, since an actin network is not necessary to observe it. A dense array of WASP proteins is.

The authors go back and forth in the paper between casting the mechanism as increasing the local density of actin near the filament end and the delivery of actin to the end by WASP proteins. Can they distinguish these possibilities? This is a subtle distinction, hinging on whether WASP, monomer and filament interact simultaneously or not during the addition step.

While the data in figure 5 appear solid, the authors' conclusion that the behavior of the PWCA construct shows that the proline-rich regions facilitate transfer of actin monomers from profilin-actin complexes is an overinterpretation. The data suggest that profilin, actin and WH2 can form a ternary complex (e.g. through binding of the beta-strand of the WH2 motif to the face of actin as observed in crystal structures, while profilin is bound to the subdomain 1-subdomain 3 cleft). But it is an

additional step to say that this represents "transfer" of actin from profilin to WH2, which implies binding and release steps and an ordered process. The authors should stay closer to the data in their description.

In the discussion the authors state that >80% of the actin is delivered by WASP family proteins in the membrane, rather than from solution. This is an interesting conclusion, and the calculation leading to it should be described.

Since the PRD construct alone can deliver profilin-actin to filament ends (cf. Figure 4), it seems reasonable that the PRD may act in trans, at least some of the time, rather than transferring actin to the WH2 domain in cis, as implied by some of the mechanistic statements in the text. The authors should couch such statements more carefully.

Some additional experiments the authors could consider to enrich the paper:

1. What is the density-dependence of the phenomenon? It would be interesting to know as point of comparison to cellular densities of WASP proteins.
2. Since free WH2 domains cap filaments, it seems that there should be an optimum rate regime of WH2 binding actin, delivering actin to a filament, and then dissociating to enable the next round of addition. This could be different for different WASP proteins. Have the authors examined the effects described for WAVE1 with any other WASP family member WCA regions?

It would be helpful if the authors could explain the difference between the behaviors they report and the behaviors reported by Khanduja and Kuhn, 2014.

Coordinate axes should be shown in panels 1B, D and F, to show that the filaments are growing in different directions relative to the surface in B versus D and F.

It is unclear whether the quantitative model described on page 9, based on a difference in rates of filament association of actin and WH2-actin, is the only way to explain the data in figure 3G. For example, could a model based on differences in dissociation rate of WH2 from actin also account for the data?

On page 10, "figure 3B,C" should be "figure 4B,C". Also, the nomenclature for PRD construct in the text and P construct in the figure should be unified.

1st Revision - authors' response

08 August 2017

Referee #1:

This is a beautiful work, which is both technically elegant and makes a major functional discovery; it conclusively demonstrates that, in addition to activating the Arp2/3 complex, WASP-family NPFs can accelerate barbed end filament elongation and link dendritic actin networks to membranes when clustered on surfaces. While the work is all in vitro, it replicates conditions that are physiologically realistic, as NPFs are often clustered on membranes. I enthusiastically recommend publication of this work in *EMBO J*, and would additionally suggest that the paper be highlighted.

We are pleased by the reviewer's positive response to the work and would be happy to help the journal highlight the work by, for example, creating additional explanatory figures.

1. My most significant concern is that the generalizing conclusion "WASP-family proteins can act as network polymerases and tethering factors" lacks experimental support. The paper only deals with one member of the WASP-family, WAVE/Scar. Therefore, at least one more NPF should be analyzed - preferably one with multiple WH2 domains. Alternatively, the conclusions should be less general.

This is a valid point, and it inspired us to perform additional experiments using the core actin-regulatory region (PWWCA) of N-WASP, which contains two WH2 sequences. These new experiments (new Figure 2G-H) reveal that N-WASP is at least as potent as WAVE1 in promoting filament elongation from profilin-actin complexes. This result argues strongly that

polymerase activity is a conserved feature of all WASP-family NPFs that contain profilin-binding, poly-proline sequences proximal to their WH2 domains. The similar polymerase activities of WAVE and N-WASP demonstrate that, when using the physiologically relevant substrate profilin-actin, the number of WH2 domains has less influence on filament elongation than the proline-rich region.

2. Add page numbers.

As frequent reviewers ourselves, we recognize that missing page numbers are annoying and make the reviewers' job harder. We apologize for this oversight, and we have added page numbers to the revised manuscript.

3. Introduction: "All of these nucleation promoting factors share a set of conserved sequences, collectively known as a WCA motif, that mediate interactions with profilin, actin, and the Arp2/3 complex" The conserved core of all NPFs is PWCA, and this is something that should probably be stated right from the start, since it is central to this study (instead of talking about PRDs in a separate paragraph, which can be harder to understand for the non-initiated).

This point is well taken and we have rewritten the Introduction to highlight the fact that the functional NPF core includes the proline rich domain.

4. Introduction: "Efficient filament formation requires simultaneous association of an Arp2/3 complex with two actin monomer-loaded WCA sequences (Gaucher et al., 2012; Padrick et al., 2008, 2011; Ti et al., 2011)." One of these references is incorrect and two are missing. Note that the two NPF model was actually disputed by Gaucher et al., 2012, and was subsequently conclusively demonstrated by Boczkowska Nat Commun. 2014 and Helgeson JBC 2014.

We appreciate the reviewer's careful reading, and especially for catching these lapses in our scholarship. We have rewritten this section of the Introduction to more accurately reflect the arguments and conclusions of the cited papers.

5. It would be preferable to add a diagram as part A of Figure 3 detailing the specific nature of all the WAVE1 constructs used in the experiments. This is now given in Supplementary Figure 2 (a simpler version can do), which is not even cited in the main text.

This is an excellent suggestion. We modified Figure 3 to include a sub-panel detailing the constructs used in our experiment and cited the panel in the main text.

6. Actually, several of the Supplemental Figures are not cited in the main text, and both 'Supplemental' and 'Supplementary' are used (not sure which is right).

We have double-checked the text of the revised manuscript to make sure that we now cite and describe all of the data in both main and EV (extended view) figures. As suggested in the manuscript formatting guidelines by EMBO J, we have tried to consolidate all previous supplemental figures in five EV figures to improve data accessibility.

7. Concerning the experiments in Figure 3 - consider disabling (as opposed to deleting) internal domains. Internal deletions change more than just the presence/absence of a domain, which can have unintended effects and obscure the conclusions.

The reviewer raises an interesting point, but we feel that it might reflect a semantic issue rather than an experimental problem. The region of WASP-family proteins responsible for activating the Arp2/3 complex and for promoting filament elongation (the PWCA region) is more or less natively unstructured in its active state. We say 'more or less' because: (1) poly-proline sequences form left-handed, type-II helices when bound to profilin and (2) WH2 domains adopt an alpha-helical structure when bound to actin or docked onto the WAVE regulatory complex. Otherwise, the entire region is a random coil, devoid of secondary or tertiary structure. Despite their lack of intrinsic secondary and tertiary structure, the proline-rich regions and WH2 sequences are often called 'domains' in the literature, erroneously

implying that they fold into stable structures. This might be the source of the reviewer's comment and so we have edited the text to make this issue clearer.

We anticipate that removing some residues could affect the spacing between binding partners but should not produce global conformational effects typically expected for globular proteins.

8. In the diagrams of Figures 4 and 5, both profilin and the PRD regions are colored yellow. Because they interact with each other, using two different colors would be preferable.

We have changed the color scheme as suggested by the reviewer.

9. Discussion: In the section entitled "The density of WASP-family proteins" it would be important to note that WASP-family proteins are subjected to activation, which could dramatically impact the actual density of these molecules that can function as polymerases. Moreover, the proteins that activate NPFs also take part of to the available membrane surface in such clusters. Finally, NPFs are often recruited to membranes through interactions with BAR domain proteins, most notably IRSp53, which are dimeric and form themselves membrane coats, with could profoundly impact the NPF polymerase mechanism proposed here.

This point is well taken. We now discuss the potential influence of activation on the density of polymerization-promoting molecules in greater detail in the revised manuscript. In response to the reviewer's comment regarding how other membrane-associated molecules (e.g. IRSp53) regulate oligomerization of WASP-family proteins, we rewrote the Discussion to describe more ways in which the polymerase activity of WASP-family proteins could be regulated by their binding partners. This is an intriguing idea that we hope to explore in the future.

Importantly, the density of WAVE1 molecules immobilized on the glass coverslips in most of our experiments (about 2,000 mCherry-NPFs/ μm^2) is actually somewhat *lower* than the best estimates from the literature of NPF density at sites of branched actin network formation *in vivo*. In the revised manuscript we now demonstrate robust polymerase activity at densities even lower than 2,000/ μm^2 (new Figure 2F). Therefore, even if only 25% of the molecules present at sites of actin assembly *in vivo* are in an active state, their density is comparable to that of the WASP-family proteins used in our experiments.

10. Discussion: Concerning the parallels between WASP-family proteins and Ena/VASP and the transfer of profilin-actin from PRDs to W sequences the authors could also check Chereau J. Struct Biol 2006 for insights.

We now cite Chereau et al. (2006) in the revised manuscript and incorporate biochemical and structural insights from this work as well as Chereau et al. (2005) into our discussion.

11. Discussion: This work demonstrates that the WH2 domain is a potent polymerase of free actin, and the proline-rich domain is a potent polymerase of profilin-actin. Therefore, it would be extremely helpful if the authors could expand on the roles of the WH2 vs. PRD regions within the context of known actin-profilin vs. actin monomer concentrations in cells.

This is a good point. Unfortunately, hard data on the concentrations of monomeric actin and profilin-actin complexes in the cytoplasm are a bit sparse. Many articles cite the Pollard, Blanchoin, and Mullins review from 2000 which, in turn, references experimental work from the 1970s (as well as some unpublished observations). These estimates all suggest that profilin-actin is the major, polymerization-competent form of actin in cells. We emphasize this point in the revised manuscript.

In general, however, the cytoskeleton community would benefit from new, live-cell measurements of the concentrations and distributions of monomeric actin and actin-profilin complexes.

12. It is also important to note that different NPFs contain different numbers of WH2 domains. How would this impact the WH2 vs. PRD interplay described above?

As noted above, we performed additional experiments comparing the distributive polymerase activity of both WAVE1 (PWCA) and N-WASP (PWWCA). In the presence of profilin-actin, we find that both NPFs accelerate single actin filament elongation rates when immobilized on patterned glass substrates (Fig. 2G-2H). The rate enhancement is remarkably similar suggesting that, in the presence of profilin, the PRD plays a more important role in setting the rate of actin filament elongation than the WH2 domains.

13. Discussion: It would be helpful if the authors could discuss the reasons for the inhibitory modulation of actin assembly by NPFs proposed by Pernier 2016 and Sweeney 2015 and comment on the differences/similarities with the stimulatory modulation proposed by Khanduja and Kuhn, 2014.

This is an excellent point. Sweeney et al. (2015) proposed that the WH2 domain of WAVE1 inhibits actin filament assembly, but this claim was based entirely on experiments with soluble and artificially dimerized NPF constructs. We find that soluble NPF fragments containing WH2 domains can, depending on the assay, indeed have two inhibitory effects on actin assembly. When measuring elongation of individual filaments they slow the rate of monomer incorporation, in a manner similar to profilin. In addition to this effect, in bulk assays of Arp2/3-stimulated actin assembly (e.g. Figure 3 of Sweeney et al., 2015) high concentrations of soluble NPFs also slow polymerization by damping spontaneous nucleation of mother filaments required for Arp2/3 nucleation activity. Neither of these effects is relevant when actin assembly takes place under more physiologically relevant boundary conditions; i.e. when NPFs are clustered on a surface.

Similar to the work presented here, Khanduja and Kuhn (2014) characterized actin filament elongation in physical contact with NPF coated surfaces. In contrast to our site-specific coupling of NPFs to substrates, however, Khanduja and Kuhn relied on non-specific binding to nanofibers. They also employ a constitutively dimeric GST-N-WASP construct for their assays, which may aid the formation of higher-order NPF assemblies upon surface adsorption. We expect that these key differences are likely resulting in (i) a considerable fraction of inactive NPF molecules and (ii) density heterogeneity and possibly local aggregation on the microneedle surface based on the following observations:

1. Khanduja and Kuhn report rapid elongation of filaments from N-WASP coated nanofibers to be rare, processive events. In addition to processive attachment, they also report frequent pausing of filament growth. Both observations might be indicative of surface heterogeneities. Unfortunately, Khanduja and Kuhn did not use a fluorescently labeled NPF in their experiments, making it impossible to correlate filament elongation with NPF protein density. However, density heterogeneity in their experiments is clearly visible from the variation in signal of NPF-bound labeled actin on the microneedle surface.

2. Khanduja and Kuhn should not have been able to reliably visualize single actin filaments on nanofibers if the surfaces were coated with physiologically relevant densities of active NPFs (1000-10000 molecules/ μm^2). Excess fluorescently labeled actin should bind the NPF surface in this case, an obstacle that prevented us from measuring actin filament dynamics by means other than using a filament-binding probe (i.e. Alexa488-UTRN) on our high-density surfaces.

3. Overall, Khanduja and Kuhn report both pausing and accelerated actin filament elongation events from N-WASP coated nanofibers. In contrast to our work, however, they never perform a molecular dissection of N-WASP to explicitly show that distinct elongation behaviors are mediated by the WH2 or other domains of N-WASP.

Pernier et al. (2016) report a variety of actin filament elongation rate measurements performed under distinctly non-physiological conditions, namely in the presence large (50-100x), super-stoichiometric concentrations of profilin relative to actin. We chose not to explore the effects of large excesses of profilin in our assay; but it is likely that we would observe similar inhibitory effects to those reported by Pernier et al. (2016).

We have added additional text to the Discussion of the revised manuscript to help clarify why both an inhibitory and stimulatory effect by NPFs have been observed by other labs.

Referee #2:

The results are solid and confer a new status to this class of proteins. They should thus be of interest to a broad readership. In particular, the observations carried out with a pattern of WAVE1 proteins are quite spectacular and convincing.

We are pleased by this reviewer's positive response to our work.

1. The surface density of WAVE1 constructs seems to be a key point, and some information should be added. How the value of 1000 molec/micron² was determined should be briefly summarized.

We agree with the reviewer that we did not sufficiently describe our methods for measuring surface density of immobilized protein in the original manuscript. We now describe these methods in much greater detail in the revised manuscript. We also performed additional measurements of the surface density of immobilized NPF molecules on PEGylated surfaces using fluorescence fluctuation spectroscopy. These additional measurements have improved our estimate of WAVE1 density, which turns out to be 1850±500 molecules per μm^2 on the patterns used in most of our experiments.

As requested by the reviewer, we also performed additional experiments to directly characterize the relationship between NPF density and filament growth rate. We find that, across the accessible density range (up to ~2,000 molecules per square micron, see Methods), filament growth velocity increases approximately linearly with density. We included these new data in new Figure 2F of the manuscript.

The authors seem to say that a low amount of WAVE1 bound to the passivated region of the coverslip could account for the observed enhancement of the elongation rate with respect to the "negative control" (Fig. 2E). Can the amount of WAVE1 bound outside the pattern be estimated (using Cy5-actin fluorescence, maybe)? It seems that this surface density would be very low, yet it would be enough to cause a 2-fold increase in elongation rate. This should be discussed.

We have analyzed the surface density of WAVE1 in the "passivated" regions of the coverslip as suggested by the reviewer and determined that these areas contain about 10% residual WAVE1 compared to high-density regions. We have added this analysis to the supplement (Supplemental Figure 3A). Notably, the growth velocity in the passivated regions is slightly higher than what could be expected from the linear density relationship we observe in independent experiments in which the surface density was systematically varied (Figure 2F). We believe that additional effects that are hard to control for, such as the degree of filament tethering (determining how proximal to the NPF-coated surfaces the barbed ends are located) might impact elongation velocities to a minor extent in our experiments. Importantly, our results indicate that even at low densities (~200 NPFs/ μm^2), the polymerase activity of NPFs is still quite potent.

For measurements on dendritic networks, were the densities on microspheres and patterns the same? It seems to me that differences in surface densities could be as important as the geometrical factors invoked by the authors. In particular, the mechanical stress induced by the convex surface of the lipid-coated microspheres is certainly reduced by the fact that the surface is fluid, thereby allowing the NPFs to be mostly on one side of the microsphere, facing the actin "comet tail".

The NPF densities in microsphere- and micropattern-based experiments differ by about one order of magnitude. Surface experiments on PEG-coated slides are performed at ~2,000 molecules/ μm^2 , while microsphere experiments approach the geometric packing limit (~15,000 NPFs/ μm^2). Because we are approaching saturating concentrations, mobility of the NPF on the bilayer does not strongly affect local density under our conditions. In line with this, we do not observe the concentration of NPF molecules on the side of the bead facing the comet tail as expected by the reviewer.

We describe these assay differences more clearly throughout the revised manuscript. Details concerning how we estimate NPF surface densities are now included in the Methods section.

Additionally, we have included more discussion and experimental data directly addressing the relationship between polymerase activity and NPF density as explained in the previous points.

NPF density differences between the two sets of experiments is unavoidable for practical reasons. Firstly, the surface immobilization method for PEGylated slides is somewhat inefficient, most likely due to the instability of PEG-maleimide groups undergoing side-reactions with water molecules during protein immobilization. Secondly, we have to leave sufficient “vacant” surface area to allow for the recruitment of streptavidin and biotin-heavy meromyosin (biotin-HMM) to tether the filaments.

2. The claim that "in the absence of actin monomers the WAVE1 WH2 domain caps and tethers filament barbed ends" (subsection title) is not supported by the data. The authors show that WH2 domains slow down filament elongation from profilin-actin but, as they mention in the end of this subsection, this could result from a dynamic competition where WH2 would transiently occupy the barbed ends (similar to the "transient and opportunistic" interaction that is proposed earlier in the manuscript).

Capping and tethering, which refer to a long-lasting bond between WH2 domains and barbed ends, cannot be excluded but are not demonstrated. In fact, they appear to be unlikely, since the rapid elongation promoted by WH2 domains in the presence of G-actin would rather require a rapid detachment of WH2 from barbed ends after a subunit has been added.

We agree with the reviewers comment that the interaction between the WH2 and barbed ends must be transient in nature. Our use of the word “caps” in the subsection title does not, however, imply that the interaction must be long-lived. To our knowledge, the terms ‘capping’ and ‘tethering’ in the actin cytoskeletal community have never been defined by specific time scales. Some readers might be familiar with the kinetics of capping protein and gelsolin, which do form long-lasting complexes with barbed ends but this is not what we observe with WH2 domains. We use “capping” and “tethering” to describe the ability of the WH2 domain to (transiently) interact with barbed ends in a manner that blocks actin monomer addition and retains them in proximity to the NPF-coated surface. This is certainly true in our work and is also consistent with previous observations from the Carlier laboratory as well as Co et al. (2007, *Cell*).

3. It is unclear to me what model the authors have in mind, exactly, to globally account for their observations. A sketch, summarizing their model for WAVE-induced accelerated elongation, would help.

We thank the reviewer for the suggestion and have added a model scheme as Figure 6 to better illustrate and explain the molecular details of NPF-stimulated filament elongation as requested.

What are the molecular details that account for the rapid elongation of filaments when they enter the WAVE1-patterned regions of the coverslip? Is the local increase in actin monomer concentration simply due to the monomers that are bound to the surface via WH2 domains, and which bind to growing barbed ends? (by the way, would this imply that WH2 rapidly detaches from its monomer after it has bound a barbed end?)

Or are these monomers in rapid exchange between the solution and the surface, thus maintaining a local increase of monomer concentration in a small volume near the surface? Additional information on these kinetics could be obtained from FRAP measurements on the surface bound Cy5-actin (or by washing out the sample with buffer).

The reviewer’s comment made us realize that we did not explain the proposed molecular mechanism as well as we should have. To be clear, we *do not* propose that WASP-family proteins increase the local concentration of *free, monomeric* actin near growing barbed ends. This would be incompatible with thermodynamic laws. Briefly, at steady state, surface-bound NPFs cannot create a monomer gradient because the rates of binding and unbinding must balance. Furthermore, a high density of free barbed ends proximal to the NPF-coated surface will act as a sink and actually *decrease* the soluble monomer concentration in this

region producing a *negative* gradient. The acceleration of filament elongation must result from transfer of monomers directly from the high-density, surface-bound pool. Our conclusion is strongly supported by our molecular dissection of WAVE1, which allowed us to pinpoint how each region of the NPF contributes to filament elongation both in *cis* and *trans*. When we describe surface-attached NPFs as increasing the local density of actin near the filament end, we are talking about this NPF-associated pool. In the revised manuscript, we have worked to make our description less ambiguous and have included a model figure to help explain the mechanism (see Figure 6).

The reviewer is absolutely correct that our model implies that the off rates of the WH2 domain and profilin from newly incorporated protomers at the barbed end of a filament must be very fast. The interaction between NPF WH2 domains and actin monomers have been studied in kinetic detail previously and are known to be transient, with published off-rates between 3-30 s⁻¹, leading to sub-second dwell times (Marchand et al 2001, Beltzner and Pollard 2007). The interactions between the NPF and monomers bound through proline-rich domains via profilin are similarly short lived. Profilin dissociates from actin monomers with a rate constant of around 1s⁻¹ (Kinosian et al. 2000), and binding of profilin-actin to individual proline-rich repeats is even more unstable with off rates exceeding 1000s⁻¹ (Vavylonis et al. 2006 and references therein).

Our own published single molecule experiments on our WAVE1 patterns confirm that monomer turnover is very rapid, with dwell times significantly shorter than 0.2 s (see Bieling et al. 2016). We therefore did not perform the FRAP or flow-out experiments suggested by the reviewer, because the time resolution of these assays is significantly lower than the previously performed single molecule assays. In conclusion, we believe that it is safe to assume that monomers bound to NPFs very rapidly turn over on NPF surfaces, regardless of whether they are bound directly to the WH2 domain or indirectly to the PRD through profilin.

Is a processive interaction with the barbed end at all possible, similar to what Richard Dickinson proposes in his models?

We have previously attempted to visualize barbed end association on Cy3-labeled WAVE1 under conditions where we can easily observe transient (~ sub-second) barbed end association of Ena/VASP proteins (Figure EV 2A-C), Hansen and Mullins, 2010 JCB). However, we were unable to observe the very transient interaction between WAVE1 (PWCA) and the elongating barbed end at the limited time resolution of this experiment, indicating that NPFs are not processive as individual molecules.

To test whether clustering of NPF molecules on surfaces can result in the processive attachment of individual filament ends, we now also carried out additional experiments on NPF-coated microspheres. These show, that in contrast to VASP, clustering cannot support processive attachment of single actin filaments to an NPF-coated surface (see new data in Figure EV2D). We made efforts to present these results more clearly in the revised manuscript.

4. The strategy for filament tethering is not clearly presented. The main text refers to biotin-streptavidin linkages, but the methods specify that HMM (which is not defined - I suppose it is heavy meromyosin) was used to anchor the filaments. Things could be made clearer by explicitly showing the filament anchoring strategy in the sketches introducing the different experiments (Figs 1A, 2A, 3A and 3E). Are the anchoring point densities the same inside and outside the WAVE1 patterns?

We apologize for this lack of clarity in our presentation. We modified the text in methods sections, figure legends, and figure diagrams to clearly state that we are using biotinylated-heavy meromyosin trapped in a rigor state and immobilized by surface-bound streptavidin molecules as a tether for all experiments in which filaments were immobilized to glass surfaces.

Other points:

a. In the caption of Figure 4C, I do not understand the last sentence "... were referenced against the PWCA all cases, g."

We have fixed this typo.

b. I find the "shadow effect" added in various 2D cartoons (e.g., around coverslip in Fig 1E, around proteins in Figs 3A, 3C, 4A, 4D, 5B-D) useless and even counter-productive in terms of clarity.

We reformatted the figure schematics removing the shadow effects to improve the clarity as suggested by the reviewer.

c. I suppose that the affinities (dissociation constants) of actin for WH2 domains are determined by the fits in Fig 5 (while profilin's affinity for the binding sites in the PRD are determined by analytical ultracentrifugation). This should be said more clearly in the main text and in the caption of Fig. 5.

We have modified the main text and figure captions to clearly state by which method affinities were determined.

Referee #3:

My main criticism is that throughout the paper the authors emphasize that dendritic actin networks grow faster than individual filaments. This occurs in the abstract, the results, the figure legends and the discussion. But unless I am missing something important, this is not what the data show.

We respectfully disagree with the reviewers comment that there is no difference in the growth velocity of dendritic actin networks growing from clusters of WASP-family proteins and isolated actin filaments incorporating monomers directly from solution. Figure 1G-1H summarize how actin networks reconstituted on lipid coated microspheres and micro-patterned glass surface grow significantly faster than single actin filaments under identical conditions (i.e. 5 μ m actin-profilin). This is an extremely important comparison because the rate constants measured for growth of isolated actin filaments from soluble monomers are used to describe and mathematically model growth of actin networks in many different contexts. In fact, all of the most widely used cell biology textbooks illustrate Arp2/3-dependent, branched actin networks as growing by the incorporation of soluble monomers. Our results demonstrate that this model cannot be correct.

Based on the reviewer's comment, however, we accept that the clarity of our presentation could be significantly improved to avoid ambiguity and/or confusion.

The data show that filaments growing on dense WAVE clusters grow faster than those growing on other substrates. The difference in filament architecture in the initial experiments--network vs single filament--is incidental to the actual mechanism. So while it may be historically true that a comparison between individual filaments growing on a PEG-coated coverslip and a dendritic network growing on a WCA-coated bead initiated the project, it is misleading to readers to introduce the work as comparing single filaments to dendritic networks.

We disagree with the reviewer's suggestion that the connection between WASP-family proteins and branched actin networks is "incidental" or "historical." The key point is that, since WASP-family proteins promote nucleation activity of the Arp2/3 complex, the substrates of their polymerase activity are always branched actin networks. In our *in vitro* experiments, dense NPF arrays *can* promote elongation of isolated filaments that are not part of a branched actin network, but this is not a physiologically relevant situation. In contrast, our observation that branched actin networks grow faster than expected, given the rate at which soluble actin monomers incorporate into free barbed ends is relevant to understanding how branched actin networks carry out a variety of biological functions.

A useful analogy might be the suggestion that MPF (Maturation/Mitosis Promoting Factor) is a misleading name because it actually refers to a kinase and its connection to mitosis and oocyte maturation is only “incidental” or “historical.” We have revised the manuscript throughout to avoid confusion. For example, we have rewritten the heading and first sentence of the Results section so that it now reads:

“Filament elongation from soluble monomers is not fast enough to account for the rate of branched actin network growth. To better understand the process of filament elongation within branched actin networks, we compared the growth of these networks to the elongation of single actin filaments in the same concentration of profilin-actin complexes.”

This comes up over and over (e.g. abstract sentence 3, legend to figure 1, start of the results, start of the discussion, etc.), and really needs to be corrected. Even the name given to the phenomenon--"network polymerase" activity--is misleading, since an actin network is not necessary to observe it. A dense array of WASP proteins is.

We take issue with the statement that “the name given to the phenomenon--‘network polymerase’ activity--is misleading, since an actin network is not necessary to observe it. A dense array of WASP proteins is.” This is akin to saying that the term “DNA polymerase” is misleading because it actually refers to a protein. Calling the collective, elongation-promoting activity of an array of NPFs a ‘network polymerase’ emphasizes the biologically relevant activity, and distinguishes it from more processive polymerases such as formins and Ena/VASP proteins, which can operate in smaller numbers and in more varied contexts. We have added additional text to the revised manuscript clarifying our use of the phrase “network polymerase.”

The authors go back and forth in the paper between casting the mechanism as increasing the local density of actin near the filament end and the delivery of actin to the end by WASP proteins. Can they distinguish these possibilities? This is a subtle distinction, hinging on whether WASP, monomer and filament interact simultaneously or not during the addition step.

The reviewer’s comment made us realize that we should have done a better job explaining our proposed molecular mechanism. To be clear, we *do not* propose that WASP-family proteins increase the local concentration of *free, monomeric* actin near growing barbed ends. This would be incompatible with thermodynamic laws. Briefly, at steady state, surface-bound NPFs cannot create a monomer gradient because the rates of binding and unbinding must balance. Furthermore, a high density of free barbed ends proximal to the NPF-coated surface will act as a sink and actually slightly *decrease* the soluble monomer concentration in this region producing a *negative* gradient. The acceleration of filament elongation must result from transfer of monomers directly from the high-density, surface-bound pool. Our conclusion is strongly supported by our molecular dissection of WAVE1, which allowed us to pinpoint how each region of the NPF contributes to filament elongation both in *cis* and *trans*. When we describe surface-attached NPFs as increasing the local density of actin near the filament end, we are talking about this NPF-associated pool. In the revised manuscript, we have worked to make our description less ambiguous and have included a model figure to help explain the mechanism (see Figure 6).

While the data in figure 5 appear solid, the authors' conclusion that the behavior of the PWCA construct shows that the proline-rich regions facilitate transfer of actin monomers from profilin-actin complexes is an overinterpretation. The data suggest that profilin, actin and WH2 can form a ternary complex (e.g. through binding of the beta-strand of the WH2 motif to the face of actin as observed in crystal structures, while profilin is bound to the subdomain 1-subdomain 3 cleft). But it is an additional step to say that this represents "transfer" of actin from profilin to WH2, which implies binding and release steps and an ordered process. The authors should stay closer to the data in their description.

We are slightly confused by this comment, and worry that our description of this experiment has confused the reviewer. Although the assay presented in Figure 5 reports only on the interaction between actin and the WH2 domain, and is not capable of detecting formation of a ternary complex, the reviewer concludes that “The data suggest that profilin, actin and WH2

can form a ternary complex.” This is a reasonable leap of logic, but we’re perplexed that the reviewer does not accept that formation of such a complex implies “transfer of actin from profilin to WH2.” In essence, the reviewer proposes that profilin-actin (PA) can form a complex with the WH2 (W), but that this PAW complex *cannot* dissociate into P+AW but must *always* return to PA+W. A simple thermodynamic analysis suggests that dissociation must proceed through both pathways, partitioned according to the ratio of the dissociation rate constants.

In the simplified energy diagram above (we are not representing the profilin/poly-proline interaction), we know the affinities of $P+A \leftrightarrow PA$ (K_{D4}) and $A+W \leftrightarrow AW$ (K_{A3}) from previous studies, and we have demonstrated that PA+W has a non-zero association constant (K_{A1}). Conservation of energy, therefore, requires that the dissociation equilibrium constant for $P+AW \leftrightarrow PAW$ (K_{D2}) cannot be zero. In other words, the dissociation of PAW into P+AW cannot be forbidden. We now include a version of this argument in the revised manuscript.

Although thermodynamic analysis indicates that the $PAW \leftrightarrow P+AW$ transition is allowed, it says nothing about the kinetics of the process. Profilin dissociates from monomeric actin with a relatively fast rate constant of 4.3 s^{-1} (Vinson et al., 1998. *Biochemistry* 37:10871), but it is formally possible that profilin dissociates much more slowly as part of the PAW complex. Crystal structures by Chereau et al. (2005), however, argue against this possibility, revealing a steric interference between profilin and the alpha helix of the WH2 domain when bound to actin. Based on this steric hindrance, we expect that profilin dissociates from the PAW complex even faster than from PA.

Finally, if both pathways for PAW dissociation are accessible, the presence of free barbed ends will promote removal of actin from AW complexes (as in Figure 2), leaving free W domain to interact with additional PA complexes. Neither profilin nor the WH2 domain has a significant effect on the critical concentration of actin assembly and so as long as profilin-actin is present above the critical concentration, the free energy of polymerization will drive an ‘ordered process’ of monomer transfer. Note that this does not imply that the *only* route of monomers onto barbed ends is via the WH2 domain. Profilin-actin can incorporate directly into barbed ends while attached to the proline-rich region. To avoid confusion, we describe our reasoning in more detail in the revised manuscript.

In the discussion the authors state that >80% of the actin is delivered by WASP family proteins in the membrane, rather than from solution. This is an interesting conclusion, and the calculation leading to it should be described.

We apologize for omitting the calculation that supports this statement from the previous version of the manuscript. We have now added a detailed description of our reasoning to the main text of the revised manuscript.

Briefly, if we assume that soluble and NPF-associated actin monomers have similar access to free barbed ends, the fraction of actin subunits that enter the network from solution depends on the relative rates of incorporation from the two pools. If we call the rates of incorporation from soluble and NPF-bound pools k and nk respectively, then the fraction of monomers incorporated from solution is simply $1/(n+1)$. A lower bound for the value of the proportionality constant, n , is simply the observed acceleration of network growth compared to the rate of elongation of an actin filament in the same concentration of profilin-actin. In our experiments, actin networks grew from micropatterned WAVE1 surfaces 3.5-fold faster than

individual filaments elongated from the same concentration of soluble profilin-actin. From this comparison we derive an upper bound on the fraction of actin incorporated from solution of 0.22.

Since the PRD construct alone can deliver profilin-actin to filament ends (cf. Figure 4), it seems reasonable that the PRD may act in *trans*, at least some of the time, rather than transferring actin to the WH2 domain in *cis*, as implied by some of the mechanistic statements in the text. The authors should couch such statements more carefully.

This is a good point and we agree that the function of different NPF domains can occur in both *cis* and *trans*. In the new model figure 6, we diagram how we imagine the polymerase activity and WH2 actin monomer loading is executed in *trans*.

Some additional experiments the authors could consider to enrich the paper:

1. What is the density-dependence of the phenomenon? It would be interesting to know as point of comparison to cellular densities of WASP proteins.

We have performed additional experiments to investigate the density dependence of NPF-assisted elongation in greater detail as suggested by the reviewer (Figure 2F).

2. Since free WH2 domains cap filaments, it seems that there should be an optimum rate regime of WH2 binding actin, delivering actin to a filament, and then dissociating to enable the next round of addition. This could be different for different WASP proteins. Have the authors examined the effects described for WAVE1 with any other WASP family member WCA regions?

This is a valid point and we have performed additional experiments using the core domain of N-WASP containing two WH2 sequences as suggested by the reviewer. These show that N-WASP (PWWCA) is similarly potent as WAVE1 in promoting filament elongation locally in the presence of profilin-actin (see new Figure 2G,H). The number of WH2 domains has only a moderate influence on elongation when profilin-actin is the major substrate. Transfer of monomers seems to be dominated by the proline-rich domain, which is similar between N-WASP and WAVE. The influence of which we consider equivalent to the situation in vivo where profilin-actin constitutes the main pool of polymerizable monomers.

It would be helpful if the authors could explain the difference between the behaviors they report and the behaviors reported by Khanduja and Kuhn, 2014.

Similar to the work presented here, Khanduja and Kuhn (2014) characterized actin filament elongation in physical contact with NPF coated surfaces. In contrast to our site-specific coupling of NPFs to substrates, however, Khanduja and Kuhn relied on non-specific binding to nanofibers. They also employ a constitutively dimeric GST-N-WASP construct for their assays, which may aid the formation of higher-order NPF assemblies upon surface adsorption. We expect that these key differences are likely resulting in (i) a considerable fraction of inactive NPF molecules and (ii) density heterogeneity and possibly local aggregation on the microneedle surface based on the following observations:

1. Khanduja and Kuhn report rapid elongation of filaments from N-WASP coated nanofibers to be rare, processive events. In addition to processive attachment, they also report frequent pausing of filament growth. Both observations might be indicative of surface heterogeneities. Unfortunately, Khanduja and Kuhn did not use a fluorescently labeled NPF in their experiments, making it impossible to correlate filament elongation with NPF protein density. However, density heterogeneity in their experiments is clearly visible from the variation in signal of NPF-bound labelled actin on the microneedle surface.

2. Khanduja and Kuhn should not have been able to reliably visualize single actin filaments on nanofibers if the fiber surfaces were coated with physiologically relevant densities of active NPFs (1000-10000 molecules/ μm^2), which bind fluorescently labeled actin monomers. This interaction prevented us from measuring actin filament dynamics on high-density NPF-coated surfaces by means other than using a filament-binding probe (i.e. Alexa488-UTRN).

3. Overall, Khanduja and Kuhn report both pausing and accelerated actin filament elongation events from N-WASP coated nanofibers. Unfortunately, they never perform a molecular dissection of N-WASP to explicitly shown that distinct filaments elongation behaviors are mediated by the WH2 domain.

Coordinate axes should be shown in panels 1B, D and F, to show that the filaments are growing in different directions relative to the surface in B versus D and F.

We added coordinate axes to Figure 1 to more clearly indicate the direction of filament growth.

It is unclear whether the quantitative model described on page 9, based on a difference in rates of filament association of actin and WH2-actin, is the only way to explain the data in figure 3G. For example, could a model based on differences in dissociation rate of WH2 from actin also account for the data?

This is a good point. While an increased dissociation rate of WH2-bound monomers from barbed ends might in principle be able to account for the observed inhibition of elongation, we experimentally verified that this is not the case (Figure EV 2E). Titration of monomers in either form – bare or bound to the WCA region of WAVE1 – reveals similar off-rates, which are not significantly different from each other.

On page 10, "figure 3B,C" should be "figure 4B,C". Also, the nomenclature for PRD construct in the text and P construct in the figure should be unified.

We made the appropriate textual changes to address the reviewers comment.

2nd Editorial Decision

16 September 2017

Thank you for submitting a revised version of your manuscript. I apologise for the delay in communicating the decision due to delayed referee reports. The manuscript has now been seen by two of the original referees, who find that their main concerns have been addressed. There remain only a few minor editorial issues that have to be resolved before formal acceptance of the manuscript.

Please let me know if you have any further questions regarding this final revision. You can use the link below to upload the revised version.

Thank you again for giving us the chance to consider your manuscript for The EMBO Journal. I am looking forward to seeing the final version.

REFeree REPORTS

Referee #1:

I am fully satisfied with the authors' thorough response, and particularly pleased that they have now added data on another NPF (N-WASP), thus making the work even more impactful. I predict the cytoskeleton community will meet this work with enthusiasm.

Referee #2:

I am satisfied with the revisions. Following my recommendations and those from the two other reviewers, the authors have improved their manuscript further.

2nd Revision - authors' response

19 September 2017

The authors made the requested changes and submitted the final version of their manuscript.

Corresponding Author Name: Peter Bieling

Journal Submitted to: EMBO J

Manuscript Number: EMBOJ-2017-97039R